J Physiol 603.7 (2025) pp 1983–2004

# Towards spatially selective efferent neuromodulation: anatomical and functional organization of cardiac fibres in the porcine cervical vagus nerve

Nicole Thompson[1] ![ORCID], Enrico Ravagli[1], Svetlana Mastitskaya[1], Ronald Challita[2], Joseph Hadaya[2], Francesco Iacoviello[3], Ahmad Shah Idil[1] ![ORCID], Paul R. Shearing[3], Olujimi A. Ajijola[2] ![ORCID], Jeffrey L. Ardell[2] ![ORCID], Kalyanam Shivkumar[2] ![ORCID], David Holder[1] ![ORCID] and Kirill Aristovich[1]

[1] EIT and Neurophysiology Research Group, Department of Medical Physics and Biomedical Engineering, University College London, London, UK
[2] UCLA Cardiac Arrhythmia Center and Neurocardiology Research Program of Excellence, David Geffen School of Medicine at UCLA, Los Angeles, California, USA
[3] Electrochemical Innovation Lab, Department of Chemical Engineering, University College London, London, UK

Handling Editors: Harold Schultz & Andrew Holmes

The peer review history is available in the Supporting Information section of this article (https://doi.org/10.1113/JP286494#support-information-section).

**Abstract** Spatially selective vagus nerve stimulation (sVNS) offers a promising approach for addressing heart disease with enhanced precision. Despite its therapeutic potential, VNS is limited by off-target effects and the need for time-consuming titration. Our research aimed to determine the spatial organization of cardiac afferent and efferent fibres within the vagus nerve of pigs to

---

N. Thompson and E. Ravagli are joint first authors.

This article was first published as a preprint. Thompson N, Ravagli E, Mastitskaya S, Challita R, Hadaya J, Iacoviello F, Idil AS, Shearing PR, Ajijola OA, Ardell JL, Shivkumar K, Holder D, Aristovich K. 2024. Anatomical and functional organization of cardiac fibers in the porcine cervical vagus nerve allows spatially selective efferent neuromodulation. bioRxiv. https://doi.org/10.1101/2024.01.09.574861

achieve targeted neuromodulation. Using trial-and-error sVNS *in vivo* and *ex vivo* micro-computed tomography fascicle tracing, we found significant spatial separation between cardiac afferent and cardiac efferent fibres at the mid-cervical level and they were localized on average on opposite sides of the nerve cross-section. This was consistent between both *in vivo* and *ex vivo* methods. Specifically, cardiac afferent fibres were located near pulmonary fibres, consistent with findings of cardiopulmonary convergent circuits and, notably, cardiac efferent fascicles were exclusive. These cardiac efferent regions were located in close proximity to the recurrent laryngeal regions. This is consistent with the roughly equitable spread across the nerve of the afferent and efferent fibres. Our study demonstrated that targeted neuromodulation via sVNS could achieve scalable heart rate decreases without eliciting cardiac afferent-related reflexes; this is desirable for reducing sympathetic overactivation associated with heart disease. These findings indicate that understanding the spatial organization of cardiac-related fibres within the vagus nerve can lead to more precise and effective VNS therapy, minimizing off-target effects and potentially mitigating the need for titration.

(Received 1 March 2024; accepted after revision 9 August 2024; first published online 24 August 2024)

**Corresponding author** N. Thompson: EIT and Neurophysiology Research Group, Department of Medical Physics and Biomedical Engineering, University College London, London, UK. Email: nicole.thompson@ucl.ac.uk

**Abstract figure legend** Functional, *in vivo* imaging was performed in pigs by means of selective vagus nerve stimulation with a multi-electrode cuff in conjunction with physiological readouts of organ-specific activity. These data were analysed and correlated to the electrode positions to determine the functional map. Subsequently, the pig was killed and the vagus nerve with its branches was dissected. Structural imaging was performed with contrast-enhanced micro-computed tomography scanning followed by 3D segmentation of the organ-specific branches up, and from the nodose ganglion down, to the mid-cervical level of the vagus nerve. Locations of the laryngeal, pulmonary and the cardiac efferent and afferent function or fascicles were identified using both the *in vivo* and *ex vivo* imaging methods, respectively, in the same animal for cross-validation. The maps show the mean organization of regions across animals ($n = 10$ *in vivo*, $n = 5$ *ex vivo*). Laryngeal fascicles were predominantly efferent and in close proximity to the cardiac efferent fascicles, and pulmonary fascicles were mostly afferent and in close proximity to the cardiac afferent fascicles that were identified to be specifically cardiopulmonary fibre-containing fascicles. Notably, the cardiac efferent and cardiac afferent regions were significantly separated and thus this gives promise for the selective neuromodulation of the cardiac efferent fascicles without activation of cardiac and pulmonary afferent-related reflexes for the treatment of heart disease.

**Key points**

- Spatially selective vagus nerve stimulation (sVNS) presents a promising approach for addressing chronic heart disease with enhanced precision.
- Our study reveals significant spatial separation between cardiac afferent and efferent fibres in the vagus nerve, particularly at the mid-cervical level.
- Utilizing trial-and-error sVNS *in vivo* and micro-computed tomography fascicle tracing, we demonstrate the potential for targeted neuromodulation, achieving therapeutic effects such as scalable heart rate decrease without stimulating cardiac afferent-related reflexes.
- This spatial understanding opens avenues for more effective VNS therapy, minimizing off-target effects and potentially eliminating the need for titration, thereby expediting therapeutic outcomes in myocardial infarction and related conditions.

## Introduction

The vagus nerve is a major focus of neuromodulation due to its extensive distribution and complex neural pathways to and from the heart, larynx, lungs and abdominal viscera. It plays a vital role in regulating various body functions. As the main peripheral nerve of the parasympathetic branch of the autonomic nervous system (ANS), it controls heart rate (HR), blood pressure, digestion, respiration and immunity. By applying electrical stimulation to the nerve, these physiological effects can either be induced or inhibited. At present, vagus nerve stimulation (VNS) is used to treat drug-resistant depression and epilepsy, but a particularly critical application for VNS is in cardiac pathologies such

as heart failure (HF), ischaemic heart disease, myocardial infarction (MI), hypertension and arrhythmia (Capilupi et al., 2020; Hadaya et al., 2023; Hanna et al., 2018; Horn et al., 2019; Li et al., 2004).

The ANS, via its vagal and sympathetic limbs, controls cardiac contractile and electrophysiologic function. Following MI, ANS dysfunction results in excessive reflex activation of efferent sympathetic tone to the heart to compensate for the reduced functional capacity of the damaged heart. MI further leads to downregulation of parasympathetic (vagal) activity; this is associated with lethal ventricular arrhythmias (Fukuda et al., 2015) and is intricately involved in the process of maladaptive cardiac remodelling (Mozaffarian et al., 2015) that ultimately leads to cardiac mortality. By performing VNS, regional cardiac electrical and mechanical function is maintained, and susceptibility of the myocardium to harmful arrhythmias and the effects of sympathetic system overactivation can be reduced thereby mitigating heart disease progression (Hadaya & Ardell, 2020; Hadaya et al., 2023; Li et al., 2004). In preclinical studies, reactive VNS has preserved autonomic control and induced cardioprotection when applied early on following MI (Hadaya et al., 2023). VNS has also shown promise for treating HF in recent clinical trials where improvements from baseline were observed in standard deviation in normal-to-normal R–R intervals, left ventricular ejection fraction and Minnesota Living with HF mean score, and an improvement in 6 min walk distance was observed (Premchand et al., 2014, 2016).

However, there are two major problems with clinical applications of VNS. First, the current paradigm of VNS is to stimulate the entire nerve with FDA-approved implantable VNS interventions that lack spatial selectivity capabilities, leading to indiscriminate stimulation of non-targeted effectors and undesired side effects (Ben-Menachem, 2001). These off-target effects are due largely to afferent fibre activation, although some side effects such as pain that could be argued to be attributed to efferent activation, have been reported, with pain reported during VNS in less than 3% of patients in the ANTHEM HF with reduced ejection fraction (HFrEF) trial (Premchand et al., 2014, 2016). Additionally, other trials that failed to demonstrate the expected benefit, namely NECTAR-HF and INOVATE-HF, are predicted to have done so due to being performed prior to a sufficient level of understanding of the anatomical organization of the vagus nerve and the complex pathophysiology of VNS with resultant delivery of therapy at sub-therapeutic levels (Dusi & De Ferrari, 2021).

Secondly, due primarily to these off-target effects on respiratory and gastrointestinal (GI) function, VNS on the whole nerve takes time (∼4 weeks) to titrate to therapeutic levels (Ardell et al., 2017; DiCarlo et al., 2017). Given that time is critical following MI, with only adverse outcomes expected from delays due to the irreversible nature of many post-MI neural and cardiac events (Furukawa et al., 1990; Hardwick et al., 2014; Jenča et al., 2021; Mastitskaya, 2023; Mechanic et al., 2023; Moyé et al., 1991; Pfeffer, 1994), it is advantageous to apply VNS as soon as possible after a heart attack, thereby reducing infarct size, preventing ANS dysfunction and improving survival.

A possible solution to overcome these hurdles is the use of spatially selective VNS (sVNS). Spatially selective VNS was first demonstrated in its modern form in the vagus nerve of sheep (Aristovich et al., 2021) for evoking separate respiratory and cardiac effects. Briefly, this work used an electrode array geometry consisting of 14 electrode pairs, placed at equally spaced angular positions around the circumference of the nerve over two electrode rings. More recently, sVNS of the vagus nerve was also reported in the pig vagus nerve using the same 14-electrode double-ring array (Thompson et al., 2023), an eight-electrode single-ring array (Jayaprakash et al., 2023) and a six-electrode FDA investigational device (Blanz et al., 2023). Selective stimulation of the right vagus nerve with multiple contacts was also achieved recently for modulation of HR (Agnesi et al., 2024); however, this work employed an invasive transversal intrafascicular multichannel electrode (TIME) array. Recently, the organization of the cervical vagus nerve in swine was investigated by Thompson et al. (2023), who showed a degree of cross-sectional organization for branches innervating the larynx, heart and lungs, which cross-correlated between sVNS, micro-computed tomography (microCT) and electrical impedance tomography (EIT). Similar conclusions were drawn

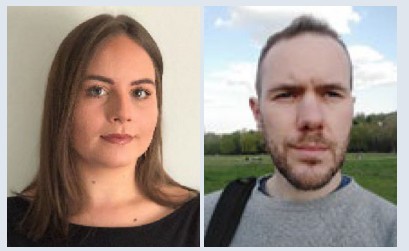

**Nicole Thompson**, a post-doctoral fellow at UCL's Neurophysiology and Electrical Impedance Tomography Research Group, holds a PhD in Neuroscience and Biomedical Engineering from UCL. Specializing in neuroimaging and neurophysiology, her research targets imaging techniques, medical devices and vagus nerve stimulation (VNS), focusing on mammalian vagus nerve anatomy She aims to enhance understanding of vagus nerve anatomy for improved neuromodulation, particularly in treating epilepsy and heart failure. She is currently coordinating a VNS clinical trial. **Enrico Ravagli** is a Senior Research Fellow at UCL's Neurophysiology and Electrical Impedance Tomography Research Group and holds a PhD in Bioengineering from the University of Bologna. His research interests include neuromodulation of the vagus nerve, electrical impedance tomography of neural tissue and developing open-source biomedical instrumentation. He is currently the holder of a UKRI fellowship and a Visiting Research at the University of Sydney, Australia.

in another recent swine study (Jayaprakash et al., 2023). However, a deeper understanding of the functional and anatomical organization of cardiac nerve fibres in the cervical vagus is needed to perform sVNS to treat ischaemic and non-ischaemic heart disease.

Neural control of cardiac function involves a multi-tiered hierarchy of interdependent reflexes, and whole nerve VNS, when delivered at the cervical level, activates both the ascending and descending projections, engaging multiple levels of the cardiac neuroaxis (Ardell et al., 2015; Hanna et al., 2018). A key to effectively and rapidly treating heart disease and achieve therapeutic efficacy is stimulating the efferent cardiac vagal fibres whilst simultaneously avoiding activating cardiac vagal afferents (Booth et al., 2021). With whole nerve VNS stimulation, the titration to mitigate off-target effects (e.g. cough and GI discomfort), takes 3–4 weeks (Ardell et al., 2017). While this reactive temporal constraint is reasonable for stage 3 HFrEF patients (Konstam et al., 2019; Premchand et al., 2014, 2016), the adverse dynamic remodelling that occurs in neural and cardiac tissues following MI is rapid and, in many cases, irreversible (Fukuda et al., 2015; Hardwick et al., 2014; Moyé et al., 1991; Pfeffer, 1994). As such, refinements in reactive VNS neuromodulation that shorten time to therapeutic efficacy increase the potential for cardioprotection (Hadaya et al., 2023; Salavatian et al., 2016).

Vagal efferent fibres regulate chronotropic, dromotropic, inotropic and lusitropic function of the heart by direct projections to heart tissues and by interacting with sympathetic projections to the heart (Hsieh et al., 1998). However, the location and organization of these respective fibre groups within the vagus nerve is unknown. A study by Settell et al. (2020) as well as the above-mentioned study by Jayaprakash et al. (2023) found that there is roughly a bimodal distribution of efferent and afferent fibres/fascicles; however, these studies were limited by either not tracing organ-specific projections or by doing so but not in the same nerve to correlate the fibre-type findings with the organ-type fascicles, that is cardiac-related fascicles.

In this study, the organization of afferent and efferent cardiac fibres in the mid-cervical porcine right vagus nerve was investigated by means of *in vivo* neuromodulation protocols and post-mortem imaging methods. The specific aims of the study were to answer the following questions:

(1) Is there spatial separation and organization of efferent and afferent cardiac fibres over the cross-section of the vagus nerve at the mid-cervical level?
(2) Is it possible to perform sVNS of efferent cardiac fibres whilst avoiding activation of cardiac afferent fibres, which would allow for improved treatment of heart disease by reducing side effects and mitigating the need for titration?

## Methods

### Ethical approval

Animal experiments were performed at UCLA. This study was performed within the guidelines set by the University of California Institutional Animal Care and Use Committee (IUCUC) and the National Institutes of Health Guide for Care and Use of Laboratory Animals. Experiments were approved by the UCLA Chancellor's Animal Research Committee with ethical approval number ARC 2016-085. The study was performed using Yorkshire castrated male pigs, weighing $53.7 \pm 3.7$ kg sourced from S&S Farms (Romana, CA, USA). The animals are housed in the vivarium with constant free access to water and fed twice a day by vivarium staff. All authors understand the ethical principles under which the journal operates and the work presented here complies with the animal ethics checklist as outlined by the journal.

### Experimental design

The study was designed with the aim of investigating the function and anatomy of the right cervical vagus and comprised an *in vivo* ($n = 10$) and *ex vivo* phase ($n = 5$). During the *in vivo* phase, sVNS of the right vagus nerve was performed with the nerve intact to assess the effect of both afferent and efferent pathways on cardiac function; sVNS was then repeated after right vagus distal vagotomy and left vagus vagotomy to isolate in steps the contribution of different neural pathways.

Additional electrophysiological recordings, namely sVNS of laryngeal and pulmonary function, were performed in the animals to collect more information about the functional anatomy of the cardiac branch in relation to other branches and fascicles.

Following completion of the electrophysiological recordings, animals were killed, and right vagus nerves were excised and subject to microCT imaging and organ-to-cervical branch tracing.

The full experimental protocol can be summarized as follows. Initially, the animal underwent anaesthesia, followed by surgical exposure of both the right and left vagus nerves, with the placement of the sVNS cuff on the right vagus nerve (Fig. 1). During spontaneous breathing without ventilation, sVNS was conducted while monitoring $ET_{CO_2}$ and breathing rate to pinpoint the pulmonary branch. EMG electrodes were strategically inserted into the right RL muscle region to facilitate identification of the RL branch through sVNS. Next, sVNS was used to assess cardiac function. Following these initial procedures, a right vagotomy was performed

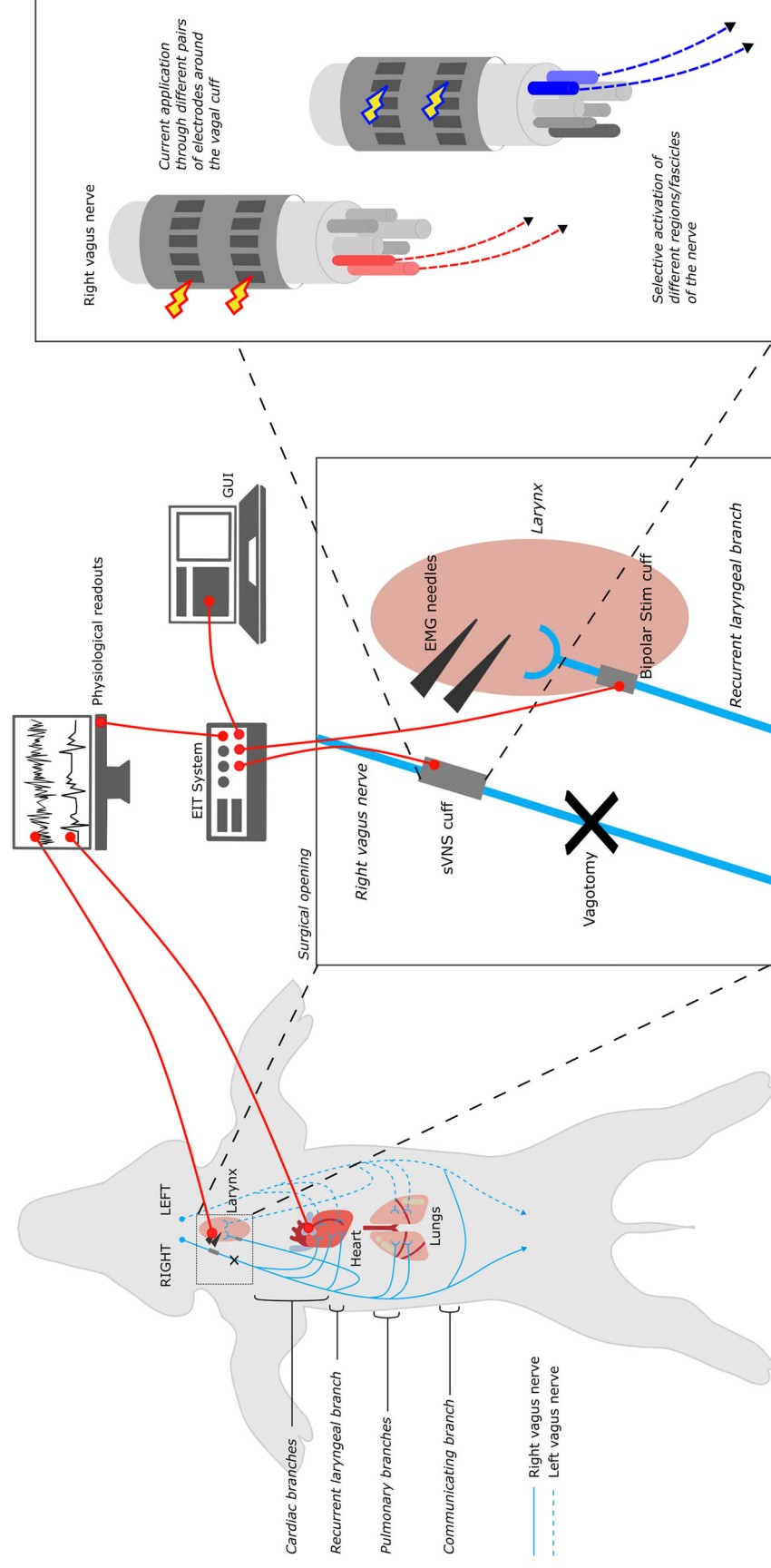

**Figure 1. Experimental *in vivo* setup (*n* = 10)**

A multi-electrode cuff for sVNS was applied to the right vagus. A custom electronic system performs sVNS on the cuff. EMG needles are placed in the larynx muscle to detect contraction during recurrent laryngeal sVNS. A physiological readout system monitors blood pressure, ECG, respiration and other vital signs. [Colour figure can be viewed at wileyonlinelibrary.com]

distal to the sVNS cuff, followed by a repeat of sVNS to evaluate cardiac function following vagotomy. Finally, a left vagotomy was performed, and again sVNS of cardiac function was repeated following vagotomy to conclude the experimental protocol.

### Surgery and anaesthesia

On the day of the experiment, Tiletamine-Zolazepam (4–6 mg/kg, .I.M.) was used for induction of anaesthesia. Animals were endotracheally intubated and mechanically ventilated (tidal volume 400–600 mL, rate 12–16 breaths/min). Anaesthesia was then maintained with isoflurane vaporized (1–4% inhaled). A continuous rate infusion of fentanyl (0.2 µg/kg/min) was started after induction and continued during subsequent surgical procedures for analgesia.

After induction of general anaesthesia, the animal was positioned in dorsal recumbency. Indwelling catheters (Millar Instruments, Inc., Houston, TX) were percutaneously placed using ultrasound guidance in both femoral arteries (for blood pressure and blood gas monitoring) and femoral veins. The animal was instrumented with a 12-lead ECG and a pulse oximeter. A spirometer was connected to the tracheal tube. The animal was mechanically ventilated using pressure control mode for the duration of the surgery and for most of the experiment, except when sVNS were applied for identification of pulmonary responses. Between these periods, if required, animals were placed onto mechanical ventilation to restore normal levels of $CO_2$ (between 35 and 45 mmHg). Body temperature was maintained using a hot air warming system if necessary. Lactate ringer fluid therapy at a rate of 5 mL/kg/h was administered intravenously throughout the procedure.

Routine anaesthesia monitoring included vital parameters such as ECG and invasive arterial blood pressure, central venous pressure, end-tidal $CO_2$ ($ET_{CO_2}$), end-tidal isoflurane ($ET_{Iso}$), pulse oximetry and core body temperature (via rectal probe). The anaesthetized animals were monitored to maintain physiological parameters within the normal or safe limits for the species, specifically: HR: 50–120 b.p.m.; mean blood pressure: 70–120 mmHg; $ET_{CO_2}$: 35–45 mmHg; and bispectral index: 50–70. Some parameters (arterial blood pressure, central venous pressure, ECG, $ET_{CO_2}$, $ET_{Iso}$) were also digitally recorded using a 16-channel PowerLab acquisition system (ADInstruments, Colorado Springs, CO, USA) with LabChart 8 software at 2 kHz sampling frequency. Levels of anaesthetic were adjusted accordingly by the anaesthetist.

The ventral neck region was clipped and aseptically prepared using chlorhexidine-based solutions. Longitudinal 20 cm skin incisions were made using monopolar electrocautery centred immediately to the left and right of the trachea, respectively. The incision was continued through the subcutaneous tissue and the sternohyoideus musculature using a sharp/blunt technique until encountering the carotid sheath and vagus nerve. A 5–7 cm segment of the left and right vagus nerves was circumferentially isolated by blunt dissection to allow placement of a sVNS electrode (right) and to allow access for vagotomy at a later stage (both nerves) (see below). The electrode cuff was placed around the right vagus nerve by carefully opening the cuff and sliding the vagus inside it, with the cuff opening facing ventrally. The sVNS cuff was placed at the mid-cervical level, ∼2–3 cm from the nodose ganglion, keeping placement of the cuffs between animals as consistent as possible. To secure the cuff in place around the nerve during the experiment, the sutures incorporated into the design for opening the cuff were tied around the cuff circumference ensuring a tight fit. Electrical ground and earth electrodes were inserted into the surgical field. The impedances of the electrodes were <1 kOhm at 1 kHz.

EMG needles (inomed Medizintechnik GmbH, Emmendingen, Germany) were implanted into laryngeal muscle on each side, specifically the cricoarytenoid and thyroarytenoid muscles, to record laryngeal effects of sVNS. The surgical field was then rinsed with sterile saline.

After the initial round of sVNS for identification of pulmonary fascicles (see 'Pre-processing, staining, microCT scanning and reconstruction' for the experimental procedure), the animal was put back on mechanical ventilation and anaesthesia was transitioned from isoflurane to alpha-chloralose (50 mg/kg initial bolus, thereafter 25–50 mg/kg/h, I.V.), an anaesthetic that does not depress cardiac and autonomic reflexes. A stabilization period of 30 min was given after anaesthesia transition. Right and left vagotomies were performed later during the experimental procedure ('Pre-processing, staining, microCT scanning and reconstruction', Fig. 2).

At the end of all recordings, an overdose of sodium pentobarbital (100 mg/kg I.V.) followed by saturated KCl (70–150 mg/kg I.V.) was used for killing. The total duration of the *in vivo* experiments from induction with anaesthesia to killing (excluding post-mortem dissection) was ∼13 h on average (Fig. 2).

### Spatially selective vagus nerve stimulation

**Instrumentation for sVNS.** All sVNS recordings in this study were performed using the ScouseTom EIT system (Avery et al., 2017). This system integrates a commercial grade EEG amplifier (Actichamp, Brain Products GmbH, Gilching, Germany), two benchtop current sources (Keithley 6221 AC, Tektronix, Brackneel, UK) and custom PCB boards for multiplexing, timing and

communication. The EEG amplifier has 24-bit resolution ($\pm$400 mV range, internal gain = 5), an internal 10 kHz hardware anti-aliasing filter, and is able to sample data in a true parallel configuration at 50,000 samples/s. Auxiliary channels from the amplifier ($\pm$5 V range, internal gain = 1), operating at the same sampling frequency and resolution, were used to acquire physiological signals ECG, $ET_{CO_2}$, laryngeal EMG, and systemic and ventricular blood pressure from the clinical monitoring system.

While this system was developed mainly for neural EIT, it can also be used for performing sVNS due to the compatible hardware features. In EIT mode, the custom multiplexer board in the system is set up to route low-amplitude sinusoidal current from one of the current sources to the nerve cuff electrode. The second current source is used to deliver current pulses to a nerve branch if the system is used for evoked EIT, or left unused if EIT of spontaneous neural traffic is being performed. In sVNS mode, the pulse-generating source is connected directly to the multiplexer board to perform sVNS, while the sinusoidal source is left on standby. For stimulation amplitudes higher than 2 mA, the current source was routed directly to the electrode array to avoid cross-talk between channels on the multiplexer board.

Electrode arrays had the same geometry as those used by Aristovich et al. (2021) and Thompson et al. (2023), and were manufactured using the same process (Chapman et al., 2019). Briefly, arrays were composed of stainless-steel tracks contained in between isolating layers of medical-grade silicone. Exposed pads were laser-patterned and coated with PEDOT:pTS to reduce contact impedance and noise from the electrode–electrolyte interface. Cuffs were glued to inner sides of silicone tubing to maintain tubular shape (2.7 mm inner diameter) for wrapping around the nerve. Arrays comprised two rings of 14 pads, each pad sized $3.00 \times 0.35$ mm$^2$. Electrodes were spaced 0.32 mm along the circumference, edge-to-edge, and rings were spaced 3.1 mm between internal edges. Two external electrodes comprising the entire cross-section of the nerve

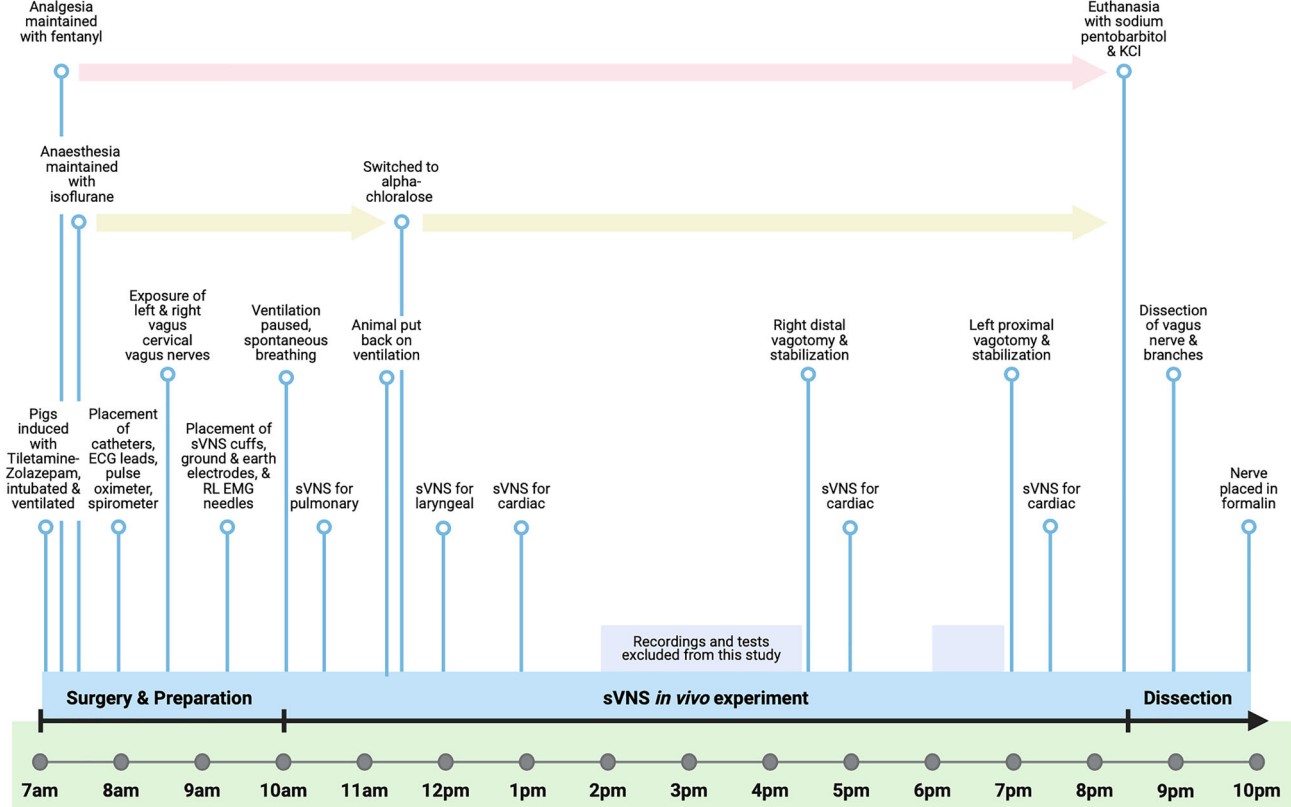

**Figure 2. Experimental *in vivo* timeline (*n* = 10)**
Experiments lasted roughly 15 h from before surgery preparation to post-mortem dissection. The *ex vivo* work was performed separately to these day experiments and are not included in this timeline. [Colour figure can be viewed at wileyonlinelibrary.com]

were present at the extremities and used as a control to perform stimulation of the full vagus nerve. These external electrodes were 0.46 mm wide, and 2.8 mm distant from the external edges of the sVNS electrodes. For electrode coordinates see Supplementary Table 1 and Supplementary Fig. 1.

**Experimental *in vivo* procedure.** Spatially selective stimulation of the right vagus nerve was performed using the above-mentioned electrode arrays embedded within circular epineural nerve cuffs. This was performed on 10 pigs ($n = 5$ recurrent laryngeal and cardiac afferent/efferent stimulation; $n = 5$ pulmonary, recurrent laryngeal and cardiac afferent/efferent stimulation).

Stimulation was applied sequentially to all 14 pairs, that is driving current through electrode pads positioned at the same angular coordinates in both electrode rings, using current biphasic pulse trains and stimulation/resting times with equal duration for each pair, e.g. 5 s on, 5 s off. For each of the target organs/functions to be identified by sVNS, a set of starting parameters was defined comprising current pulse amplitude, width and repetition frequency. These parameters were applied for the first round of trial-and-error sVNS and modified until target response was observed on less than 50% of the electrode pairs, that is 7 out of 14.

Improvement of selectivity was mostly performed by adjusting pulse amplitude but in a limited amount of cases pulse width was subject to adjustment to deliver a larger amount of charge to the nerve without compromising the electronics with large current amplitudes.

The main target function was cardiac and was monitored by recording and displaying in real time the variations in HR during the trials. Additional target functions were presence/absence of laryngeal contraction, monitored by recording laryngeal EMG, and pulmonary function, monitored by recording breathing rate during periods of spontaneous breathing. These three functions were overall assessed similarly to a previous study which performed trial-end-error sVNS on the same organs.

Pulmonary sVNS was performed first, whilst the animal was still on isoflurane. During a stabilization period (30 min) after anaesthesia transition, rounds of sVNS were performed to identify the recurrent laryngeal fascicles by observing and recording laryngeal EMG response during stimulation. Afterwards, selective stimulation for identification of cardiac efferent activity (pre-vagotomy) was performed. A right vagotomy was performed ∼1–2 cm distal to the sVNS cuff and after another stabilization period, further sVNS performed on the right vagus nerve to achieve cardiac afferent activation. Subsequently, a left vagotomy was performed on the left vagus nerve, followed by further cardiac afferent sVNS.

This concluded the experimental procedure. See Fig. 2 for a detailed timeline.

Starting parameters for each function were:

- Pulmonary – all pairs, 0.8 mA, 50 μs, 20 Hz, 15 s on/off ($n = 4/5$).
- Laryngeal – all pairs, 0.2 mA, 50 μs, 20 Hz, 5 s on/off ($n = 9/10$).
- Cardiac (efferent, pre-vagotomy) – all pairs, 1 mA, 1 ms, 10 Hz, 15 s on/off ($n = 10$).
- Cardiac (afferent, post-vagotomy) – every 2nd pair, 5 mA, 2 ms, 10 Hz, 15 s on/up to 1 min off ($n = 10$).

## Post-mortem dissection, microCT imaging and segmentation of vagus nerve fascicles

**Post-mortem dissection.** Following killing, the sVNS cuffs were sutured to the respective nerves to ensure maintenance of position during dissection. The left and right vagus nerves were then dissected from above the nodose ganglion, across the vagotomy, and down to beyond the pulmonary branches (Fig. 3). All branches exiting the vagus within this length of the nerve were traced to their originating end-organ to verify branch type which included multiple cardiac branches (including the superior and inferior branches, and smaller branches identified near the branching point of the recurrent laryngeal branch), the recurrent laryngeal branch and a few pulmonary branches. Each sample was ∼28 cm in length. Sutures were placed around the vagus nerve proximal to branching regions and to mark the cuff positions. The nerves were then fixed in 4% paraformaldehyde at 4°C for at least 24 h.

**Pre-processing, staining, microCT scanning and reconstruction.** After fixation, nerve samples were measured and sutures superglued to the vagal trunk at 4.5 cm intervals, and nerves cut into 4.5 cm lengths at the level of suture placement leaving half of the suture on the end of each section as a marker for subsequent co-registration. Two to three sections were placed into a tube of 50 mL Lugol's solution (total iodine 1%; 0.74% KI, 0.37% I) (Sigma Aldrich L6141, St Louis, MO, USA) for 5 days prior to scanning to achieve maximum contrast between fascicles and the rest of the nerve tissue. On the day of the microCT scan, the nerve segments were removed from the tube, blotted dry, ordered and wrapped in cling film (10 × 5 cm) to retain moisture during the scan. The sealed nerve samples were rolled around a cylinder of sponge (0.5 × 5 cm) and wrapped in another two layers of cling film to form a tightly wound cylinder with a diameter of ∼1.5 cm to fit within the field of view at the required resolution. The wrapped cylinder was placed inside a 3D-printed mount filled with sponge around the edges, ensuring a tight fit.

A microCT scanner (Nikon XT H 225, Nikon Metrology, Tring, UK) was homed and then conditioned at 200 kVp for 10 minbefore scanning and the target changed to molybdenum. The scanning parameters were as follows: 35 kVp energy, 120 μA current, 7 W power, an exposure of 0.25 fps, optimized projections and a resolution with isotropic voxel size of 7 μm. Scans were reconstructed in CT Pro 3D (Nikon's software for reconstructing CT data generated by Nikon Metrology). The centre of rotation was calculated manually with dual slice selection. Beam hardening correction was performed with a preset of 2 and coefficient of 0.0. The reconstructions were saved as TIFF 16-bit image stack files allowing for subsequent image analysis and segmentation in various software.

**Image and nerve analysis, segmentation and tracing.** Reconstructed microCT scan images were analysed in ImageJ in the XY plane to view the cross-section of the nerve and AVI files were created which allowed for validation of the scanning protocol, visual analysis of the

quality of the image and the distinguishability of the soft tissue, and enabled stack slice evaluation, identification of suture positions and branching locations of the vagus nerve and served as a reference during segmentation. This was performed on $n = 5$ right vagus nerves (nerves 6–10 of $n = 10$ sVNS nerves – see Supplementary Information), and these nerves were from the animals in which the *in vivo* sVNS experiment was performed to allow for accurate cross-validation of functional and structural imaging within the same animal.

Image stacks (XY plane along the Z-axis) were loaded into Vesselucida 360 (Version 2021.1.3, MBF Bioscience LLC, Williston, VT, USA) and image histograms were adjusted to optimize visualization of the fascicles when required. Starting from fascicle identification within branches of the vagus nerve, the fascicles were traced through the XY image stack of each scan up and through each cross-section to the cervical region at the mid-level of cuff placement using the Vessel mode from the Trace toolkit. Seed points were placed in the centre of the fascicle of interest along the length of the nerve at regular

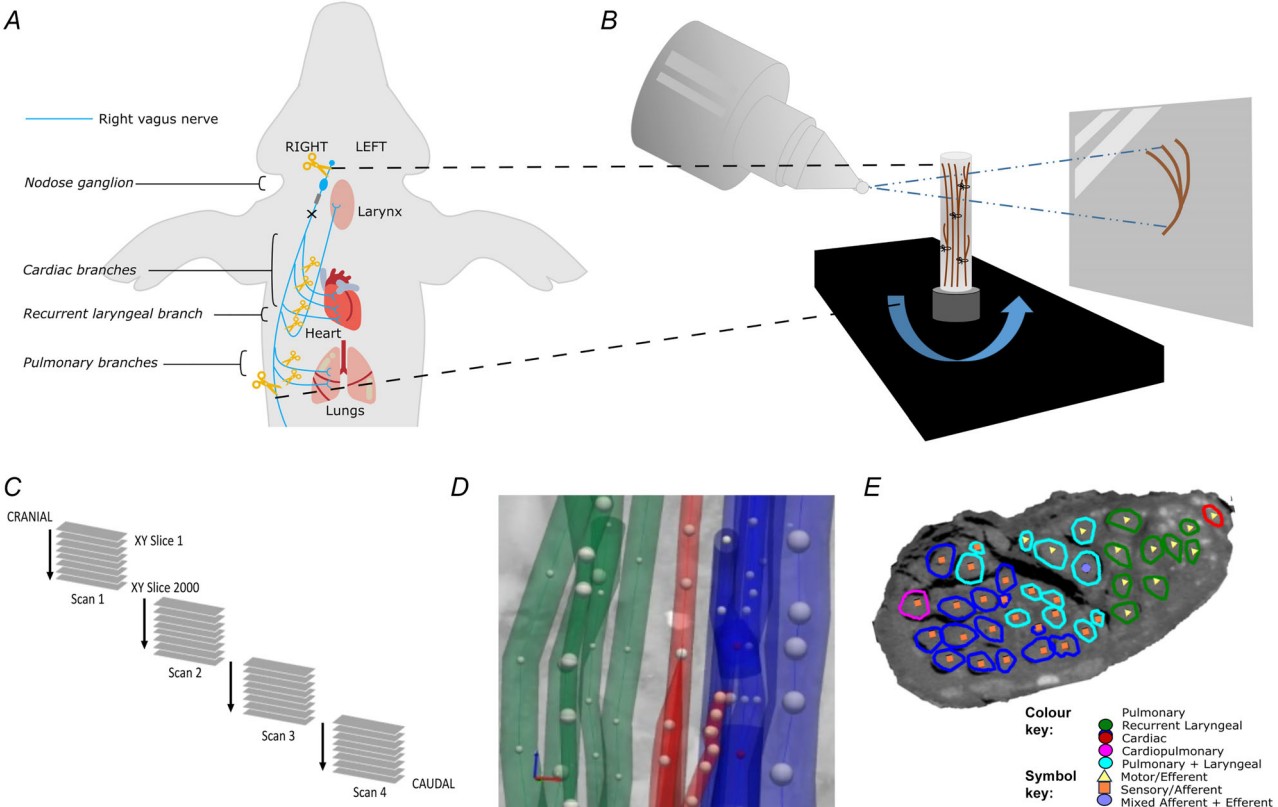

**Figure 3. MicroCT *ex vivo* imaging**
*A*, the right vagus of the pig is dissected post-mortem. The length of the nerve from above the nodose ganglion to below the pulmonary branches is dissected along with ∼1 cm of each branch. *B*, the nerve is contrast-stained, sectioned and labelled, and microCT scanned. *C*, multiple, overlapping scans are required to image the full length of the bundle of nerve sections. *D*, the fascicles are traced from their organ origin to the cervical level by placing seed points along their path. *E*, the result is a labelled and mapped cross-section at the mid-cervical level. [Colour figure can be viewed at wileyonlinelibrary.com]

cross-sections whilst adjusting the diameter to match the fascicle size, which ultimately created a 3D segmentation of fascicles labelled according to their organ origin. Bi- and trifurcations were created when fascicles split into two or more fascicles and markers placed when fascicles merged. When fascicles labelled as containing fibres of a certain type merged with another type, the subsequent fascicle was labelled as mixed, containing fibres from both. The same process was repeated but starting at the nodose ganglion and proceeding down the nerve to mid-cervical level labelling the fascicles that bypass the nodose ganglion as efferent/motor and those that originate from the nodose ganglion as afferent/sensory. Fascicles were segmented in two dimensions from the rest of the nerve using the Contour mode from the Trace toolkit by forming a closed loop around the boundary of the fascicle of interest at the start or end of each scan (depending on direction of tracing) and in the mid-cuff level for visualization of the labelled fascicles in a cross-section. The suture landmarks placed prior to cutting the nerve into segments were used to match up the neighbouring cross-sections; to continue tracing across cut regions of the nerve, the super-glued suture markers and distinct physiological regions or landmarks were used to align the proximal and distal ends of the cut nerves and tracing continued.

Subsequent to tracing of fascicles from both the proximal and distal ends, the nerves were analysed for the number and type of fascicles at cervical level (see Supplementary Information).

## Statistical analysis

See Supplementary Information for all microCT and electrophysiological data per nerve. MicroCT data were co-registered to the circle by computing the distances from each fascicle to each of the 14 electrodes and projecting them onto the circular cross-section such that radial distances to the nearest electrode were maintained. Then, the area occupied by each fascicle was assigned 1, whilst remaining area was assigned 0. The resulting maps were also rotated such that the cardiac efferent fascicle was located at the top at 0° for each pig ($n = 20$ in $N = 5$ animals, $n = 4$ per animal).

The total 'atlas' maps were computed by averaging the map for each group across all animals and normalizing it to 1. This resulted in four maps of fascicular presence across all animals, where 0 indicated that no fascicles of a particular type were present, and 1 meant that all animals had a fascicle of a particular type in a given location ($N = 5$ animals for each map).

The physiological responses recorded during selective stimulation trials were processed using back-projection mapping. First, the circular nerve cross-section was sub-divided onto 14 sectors, corresponding to each of the

stimulating pairs. The physiological variations were: HR change (%) with respect to the baseline for cardiac efferent and afferent, $ET_{CO_2}$ change (%) with respect to the baseline for respiratory, and RMS EMG change (%) with respect to the baseline for laryngeal activity, were projected on every pixel within the sector of the circle for each recording in each pig ($n = 40$ recordings in $N = 10$ animals). The four maps (recurrent laryngeal, pulmonary, cardiac afferent, cardiac efferent) were then rotated such that the cardiac efferent area was at the top of the cross-section at 0° for each pig.

Similar to microCT, the 'atlas' maps were computed by averaging the map for each group across all animals and normalizing it to 1, resulting in four total summary maps across animals ($N = 10$ animals for each map).

The average fascicular areas (%) to the total nerve cross-section for each fascicle group and technique were computed, as well as the area of overlap (%) to the corresponding fascicular areas.

Centres of mass (CoMs) were computed for each fascicular group's map for each nerve and each technique ($n = 40$ in $N = 10$ pigs for electrophysiology, $n = 20$ in $N = 5$ pigs for microCT), and the angular locations were used for ANOVA with multiple comparison tests to compute the statistical significance of group localizations, and these statistical tests were uniform throughout. This resulted in the assessment of whether the groups of fascicles/activation areas were significantly different from each other (Pulmonary *vs.* Laryngeal, Laryngeal *vs.* Cardiac Efferent, Cardiac Efferent *vs.* Cardiac Afferent, Cardiac Afferent *vs.* Laryngeal, Pulmonary *vs.* Cardiac Efferent, Pulmonary *vs.* Cardiac Efferent) for each technique (microCT *vs.* sVNS), and if the identified locations are similar between the techniques.

## Results

### Modulation of heart rate by sVNS

Results are reported as a mean ± SD. There were $2.0 \pm 1.7$ most effective electrode pairs, defined as pairs eliciting at least 75% of maximal response, out of 14 that elicited an HR decrease at the optimal current level, which induced a $-7.8 \pm 3.4\%$ HR change, prior to vagotomy (see Fig. 4 for example raw data per organ correlated to structural imaging; see Supplementary Information for raw data). For breathing rate (BR), the most effective pairs were $2.0 \pm 0.8$ out of 14, with an induced BR change of $-73 \pm 21\%$. After distal vagotomy of RV, HR variations were identified by sVNS in all animals, either a decrease of $-5.0 \pm 3.1\%$ ($n = 4/10$), disappearing after LV distal vagotomy, or an increase of $10.2 \pm 5.3\%$ ($n = 6/10$), sustained after LV vagotomy.

The parameters of the current required to achieve a spatially selective response were $1.2 \pm 0.6$ mA

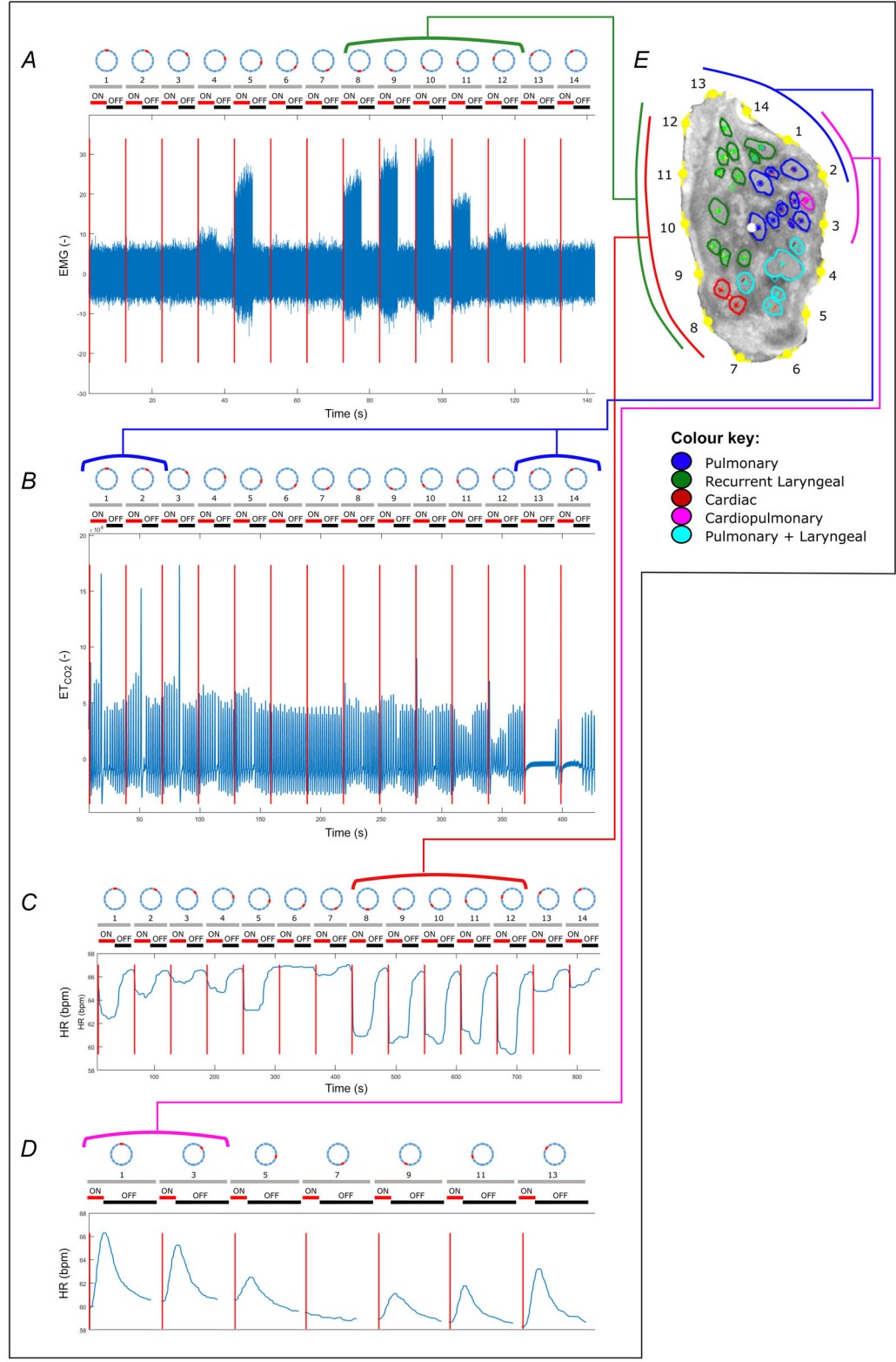

**Figure 4. An example (*n* = 1/10) of the correlation between the structural and functional imaging of the porcine vagus nerve of laryngeal, pulmonary, cardiac efferent and cardiac afferent function**

The structurally imaged, and microCT traced cross-section (*E*) of the vagus nerve with electrode positions (1–14) is compared to the electrophysiological recordings of: *A*, laryngeal EMG; *B*, pulmonary $ET_{CO_2}$; *C*, heart rate before vagotomy; and *D*, heart rate following vagotomy, during selective stimulation of the vagus nerve. The grey, numbered bars above each of the four traces represent the electrode positions on the sVNS cuff and the red and

black bars represent the stimulation condition of on and off, respectively. Stimulation periods were 5 s on/off for *A*, 15 s on/off for *B* and *C*, and 15 s on/up to 1 min off for *D*. [Colour figure can be viewed at wileyonlinelibrary.com]

amplitude, 0.5–2.0 mA amplitude and 0.1–4.0 ms pulse duration for activating cardiac efferent, while it was 5 ± 3 mA, 2 ms for activating cardiac afferent fibres. For pulmonary and laryngeal selective response, the parameters were 0.7–2.0 mA amplitude, 50–700 μs pulse duration and 150–400 μA amplitude, 50–100 μs pulse duration, respectively. A supramaximal laryngeal response was observed in every single pair for every pig while stimulating with parameters required for selective cardiac effects (all amplitudes above 1 mA).

## Composition of the vagus nerve at mid-cervical level determined by microCT tracing

The vagus nerves at the mid-cervical, mid-cuff level had a diameter of 6.63 ± 0.6 mm and an area of 2.68 ± 0.6 mm$^2$. There were 1.2 ± 0.5 cardiac, 10.2 ± 1.8 recurrent laryngeal, 10.4 ± 1.9 pulmonary, 1.4 ± 0.6 cardiopulmonary and 6 ± 2 laryngopulmonary fibre-containing fascicles (total of 29.2 ± 2.2 fascicles per nerve). Of these, 13.6 ± 3.7 (47%) were identified as afferent/sensory fascicles (defined in this study as those fascicles originating from the nodose ganglion), 10.6 ± 1.1 (36%) fascicles were identified as primarily efferent, or as those fascicles that had bypassed the nodose ganglion (henceforth referred to as efferent), and 5 ± 3.3 (17%) fascicles were identified as containing mixed fibres. The fascicles identified as cardiac were those originating from the superior and inferior cardiac branches; these did not mix with any other fascicles within the vagus nerve. From all the nerves investigated, 100% of the purely cardiac fascicles contained only efferent fibres. Both the pulmonary and cardiopulmonary fibre-containing fascicles were identified as mostly afferent with 88.5 ± 33.6% afferent and 9.6 ± 13.6% mixed fibres, and 71.4 ± 50.51% afferent and 28.6 ± 63.9% mixed fibres, respectively. There were zero purely efferent pulmonary or cardiopulmonary fascicles. Thus, afferent projections are mixed with inputs from the heart and the lungs. Laryngeal fascicles consisted of 82.4 ± 8.8% efferent, 17.7 ± 24.4% mixed fibres and 0% afferent only fascicles. Those fascicles that contained fibres originating from both the lungs and the larynx were 56.7 ± 13.9% afferent, 13.3 ± 13.9% efferent and 30.0 ± 34.2% mixed (Figs 5 and 6; Table 1).

## Organization of afferent and efferent cardiac fibres in the porcine mid-cervical vagus nerve

Fascicles referred to as cardiopulmonary, identified in microCT tracing as containing fibres from both the heart and lungs, are also referred to here as cardiac afferent as they contain fibres originating from the nodose ganglion (afferent) and those going to the heart (cardiac). This corresponds with those areas of the nerves referred to as cardiac afferent from the selective stimulation data.

Co-registration of CoMs from the microCT data ($n = 5$) of the four organ-specific regions, including laryngeal (L), pulmonary (P), cardiac efferent (CE) and cardiac afferent (CA), showed significantly different locations between all regions within the cross-section of the nerve (L-P $P = 0.008$, L-CE $P = 0.0003$, L-CA $P = 0.009$, P-CE $P = 0.005$, CE-CA $P = 0.0003$, Table 2), except between cardiac afferent and pulmonary ($P = 0.152$, Table 2). This is understandable due to the merger of cardiac with pulmonary fascicles, forming cardiopulmonary afferent fibre-containing fascicles. Fascicles of afferent *vs.* efferent nature appear to co-locate in a bimodal manner. Analysis of the CoMs from the electrophysiology data ($n = 10$) showed significant separation (L-P $P = 0.016$, L-CA $P = 0.002$, P-CE $P = 0.017$, P-CA $P = 0.019$, CE-CA $P = 0.002$, Table 2) between all regions of the vagus nerve cross-section except for cardiac efferent and laryngeal activity ($P = 0.387$) (Fig. 7; Table 2). Between the *in vivo* and *ex vivo* techniques, the location of the cardiac efferent, cardiac afferent and pulmonary regions within the nerve was consistent, that is there was no significant difference in the location of the CoMs (P $P = 0.239$, CE $P = 0.184$, CA $P = 0.195$, Table 2). However, the laryngeal regions were located in significantly different areas of the nerve as determined by microCT and selective stimulation ($P = 0.004$, Table 2). This discrepancy was probably caused by large variability and spread of laryngeal fascicles across the nerve, distortions caused by the preservation preparations and the dominant presence of connective tissue around that area of the array.

Notably, significantly different angular locations of the CoMs were identified for the cardiac afferent and cardiac efferent regions by microCT tracing (179 ± 55° between CoMs around the centre point of the nerve circular cross-section, $P = 0.0003$) and the post- *vs.* pre-vagotomy cardiac neuromodulation sites identified selective stimulation (200 ± 137°, $P = 0.002$).

From the fraction of the total area of nerve occupied by the respective regions identified by the two imaging techniques, there was a 0% overlap between cardiac afferent and efferent regions, a 93% overlap of cardiac afferent with pulmonary, 48% overlap of pulmonary with cardiac afferent, and only a 30% and 10% overlap of cardiac efferent with pulmonary and pulmonary with cardiac efferent, respectively, identified with microCT tracing (Table 2). With the electrophysiology data, the fractions of the areas of the nerve that overlapped

**Table 1. Quantitative assessment of the vagus nerve cross-section and proportions and counts of organ- and fibre type-specific fascicles of the mid-cervical vagus nerve from microCT tracing data**

| | | Nerve | | | | | Mean | SD | |
|---|---|---|---|---|---|---|---|---|---|
| | | 6 | 7 | 8 | 9 | 10 | | | |
| | Perimeter (mm) | 6.09 | 6.85 | 6.06 | 7.56 | 6.60 | 6.63 | 0.62 | |
| | Area (mm$^2$) | 2.03 | 3.07 | 2.28 | 3.40 | 2.63 | 2.68 | 0.56 | |

| | | Nerve | | | | | Mean | SD | % Total |
|---|---|---|---|---|---|---|---|---|---|
| | | 6 | 7 | 8 | 9 | 10 | | | |
| No. of | Aff/Sensory | 14 | 14 | 19 | 12 | 9 | 13.60 | 3.65 | 46.58 |
| fascicles | Eff/Motor | 11 | 9 | 12 | 11 | 10 | 10.60 | 1.14 | 36.30 |
| | Mixed | 2 | 6 | 2 | 5 | 10 | 5.00 | 3.32 | 17.12 |
| | Total | 27 | 29 | 33 | 28 | 29 | 29.20 | 2.28 | |

| | | Nerve | | | | | Mean | SD | % Total |
|---|---|---|---|---|---|---|---|---|---|
| | | 6 | 7 | 8 | 9 | 10 | | | |
| No. of | C | 1 | 1 | 1 | 2 | 1 | 1.20 | 0.45 | 4.11 |
| fascicles | L | 9 | 9 | 9 | 11 | 13 | 10.20 | 1.79 | 34.93 |
| | P | 11 | 11 | 13 | 9 | 8 | 10.40 | 1.95 | 35.62 |
| | CP | 2 | 1 | 1 | 1 | 2 | 1.40 | 0.55 | 4.79 |
| | P+L | 4 | 7 | 9 | 5 | 5 | 6.00 | 2.00 | 20.55 |
| | Total | 27 | 29 | 33 | 28 | 29 | 29.20 | 2.28 | |

| | | Nerve | | | | | Mean | SD | % Total | % Type | SD % Type |
|---|---|---|---|---|---|---|---|---|---|---|---|
| | | 6 | 7 | 8 | 9 | 10 | | | | | |
| No. of | C | 1 | 1 | 1 | 2 | 1 | 1.20 | 0.45 | 4.11 | 100.00 | |
| fascicles | C - Aff | 0 | 0 | 0 | 0 | 0 | 0.00 | 0.00 | 0.00 | 0.00 | 0.00 |
| | C - Eff | 1 | 1 | 1 | 2 | 1 | 1.20 | 0.45 | 4.11 | 100.00 | 37.27 |
| | C - Mixed | 0 | 0 | 0 | 0 | 0 | 0.00 | 0.00 | 0.00 | 0.00 | 0.00 |
| | L | 9 | 9 | 9 | 11 | 13 | 10.20 | 1.79 | 34.93 | 100.00 | |
| | L - Aff | 0 | 0 | 0 | 0 | 0 | 0.00 | 0.00 | 0.00 | 0.00 | 0.00 |
| | L - Eff | 9 | 8 | 9 | 9 | 7 | 8.40 | 0.89 | 28.77 | 82.35 | 8.77 |
| | L - Mixed | 0 | 1 | 0 | 2 | 6 | 1.80 | 2.49 | 6.16 | 17.65 | 24.41 |
| | P | 11 | 11 | 13 | 9 | 8 | 10.40 | 1.95 | 35.62 | 100.00 | |
| | P - Aff | 11 | 11 | 13 | 6 | 5 | 9.20 | 3.49 | 31.51 | 88.46 | 33.59 |
| | P - Eff | 0 | 0 | 0 | 0 | 0 | 0.00 | 0.00 | 0.00 | 0.00 | 0.00 |
| | P - Mixed | 0 | 0 | 0 | 3 | 2 | 1.00 | 1.41 | 3.42 | 9.62 | 13.60 |
| | CP | 2 | 1 | 1 | 1 | 2 | 1.40 | 0.55 | 4.79 | 100.00 | |
| | CP - Aff | 2 | 1 | 1 | 1 | 0 | 1.00 | 0.71 | 3.42 | 71.43 | 50.51 |
| | CP - Eff | 0 | 0 | 0 | 0 | 0 | 0.00 | 0.00 | 0.00 | 0.00 | 0.00 |
| | CP - Mixed | 0 | 0 | 0 | 0 | 2 | 0.40 | 0.89 | 1.37 | 28.57 | 63.89 |
| | P+L | 4 | 7 | 9 | 5 | 5 | 6.00 | 2.00 | 20.55 | 100.00 | |
| | P+L - Aff | 1 | 2 | 5 | 5 | 4 | 3.40 | 1.82 | 11.64 | 56.67 | 30.28 |
| | P+L - Eff | 1 | 0 | 2 | 0 | 1 | 0.80 | 0.84 | 2.74 | 13.33 | 13.94 |
| | P+L - Mixed | 2 | 5 | 2 | 0 | 0 | 1.80 | 2.05 | 6.16 | 30.00 | 34.16 |

L = laryngeal (green), P = pulmonary (blue), C = cardiac (red), CP = cardiopulmonary (pink), Aff = afferent (orange), Eff = efferent (yellow), Mixed (purple). SD = Standard deviation.

between cardiac afferent and efferent, cardiac afferent and pulmonary, and cardiac efferent and pulmonary were 43/82%, 78/96% and 79/52% vice versa, respectively.

## Discussion

Overall, there was a significant spatial separation of the cardiac afferent and the cardiac efferent regions within the cross-section of the mid-cervical vagus nerve at the level of VNS cuff placement. This was shown with both *in vivo* and *ex vivo* methods. Fascicles identified with *ex vivo* microCT tracing as originating from the superior and inferior cardiac branches of the vagus nerve and also identified to be fascicles that bypassed the nodose ganglion were on average on the opposite side of the nerve of fascicles identified as containing afferent (stemming from the nodose ganglion) fibres of both cardiac and pulmonary origin. In addition, cardiac recordings from the vagus nerve *in vivo* before (efferent) and after (afferent) vagotomy identified significantly separated neuromodulation sites. The location of the cardiac afferent region appeared to be predominantly located within or near the pulmonary region of the

nerve. The cardiac efferent regions were located in close proximity to the recurrent laryngeal regions. This is consistent with the roughly equitable spread across the nerve of the afferent and efferent fibres. The right vagus nerve was prioritized in this study as it is hypothesized that cardiac effects, including bradycardia and asystole, are mediated primarily by the right vagus (Howland, 2014).

The afferent effect, determined subsequent to right distal vagotomy and thus eliminating the possibility of right efferent activity, was different across the 10 animals. Activation of the afferent fibres of the right vagus nerve resulted in either tachycardia ($n = 6$) or bradycardia ($n = 4$) (Fig. 8). The possibility of vagal influence on the heart via reflexively activated efferent fibres of the left vagus during stimulation of right afferent fibres was ruled out in the control recordings with right afferent vagus stimulation before and after left vagotomy. If tachycardia was achieved during afferent stimulation, this persisted subsequent to left vagotomy, suggesting a reflex activation of the sympathetic chain. If bradycardia was the afferent effect achieved upon right vagus nerve stimulation, this effect was eliminated subsequent to left vagotomy, confirming the effect was due to activation of

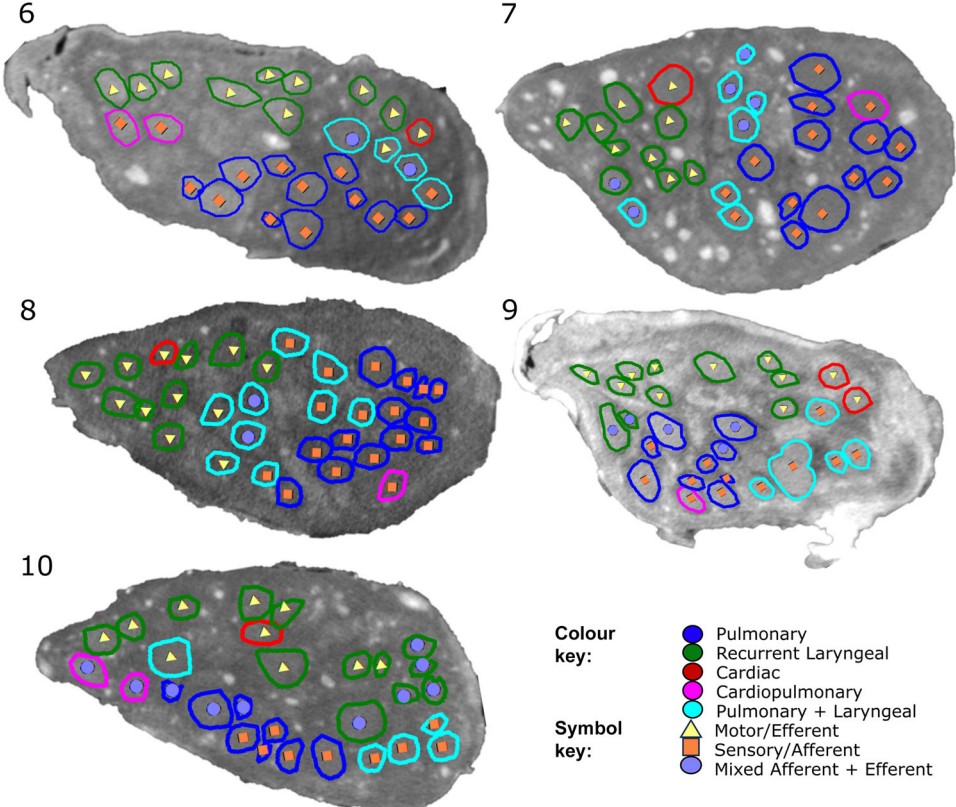

**Figure 5. MicroCT traced and labelled nerve cross-sections at the mid-cervical level for *n* = 5 right vagus nerves**
The fascicles were traced from the three target end-organs up to the mid-cervical level to decipher the organ-specific organization as well as down from the nodose ganglion to mid-cervical level to determine the distribution of afferent and efferent fibres. [Colour figure can be viewed at wileyonlinelibrary.com]

**Table 2. Calculations of the area of nerve and overlap between identified nerve regions within and between techniques**

Fraction of total area of the nerve

|  | MicroCT | | | | sVNS | | | | Between |
|  | L | P | CE | CA | L | P | CE | CA | techniques |
|---|---|---|---|---|---|---|---|---|---|
| L | 0.74 | 0.42 | 0.23 | 0.24 | 0.43 | 0.40 | 0.32 | 0.40 | 0.40 |
| P | 0.42 | 0.65 | 0.07 | 0.31 | 0.40 | 0.61 | 0.32 | 0.59 | 0.50 |
| CE | 0.23 | 0.07 | 0.23 | 0.00 | 0.32 | 0.32 | 0.40 | 0.33 | 0.22 |
| CA | 0.24 | 0.31 | 0.00 | 0.34 | 0.40 | 0.59 | 0.33 | 0.75 | 0.33 |

Percentage overlap

|  | MicroCT | sVNS |
|---|---|---|
| CE with CA | 0.00 | 82.13 |
| CA with CE | 0.00 | 43.82 |
| CA with P | 93.70 | 77.99 |
| P with CA | 48.45 | 95.56 |
| CE with P | 30.21 | 79.09 |
| P with CE | 10.61 | 51.70 |

Angles for centres of mass (°)

|  | MicroCT | | sVNS | |
|  | Mean | SD | Mean | SD |
|---|---|---|---|---|
| L | 266.09 | 60.9 | −6.22 | 28.7 |
| P | 108.03 | 64.4 | 74.35 | 72.8 |
| CE | -33.00 | 23.1 | −10.52 | 42.7 |
| CA | 145.64 | 32.2 | 189.40 | 94.5 |

Angles between centres of mass (°)

|  | MicroCT | | sVNS | |
|  | Mean | SD | Mean | SD |
|---|---|---|---|---|
| CE-CA | 179.00 | 55.30 | 200.00 | 137.20 |
| CE-P | 141.03 | 87.50 | 85.35 | 115.50 |
| CA-P | 37.61 | 96.60 | 115.05 | 167.30 |

*P* values of difference between centres of mass

|  | MicroCT | | | | sVNS | | | | Between |
|  | L | P | CE | CA | L | P | CE | CA | techniques |
|---|---|---|---|---|---|---|---|---|---|
| L | 0.5000 | 0.0082 | 0.0003 | 0.0088 | 0.5000 | 0.0159 | 0.3870 | 0.0017 | 0.0004 |
| P | 0.0082 | 0.5000 | 0.0050 | 0.1517 | 0.0159 | 0.5000 | 0.0167 | 0.0190 | 0.2389 |
| CE | 0.0003 | 0.0050 | 0.5000 | 0.0003 | 0.3870 | 0.0167 | 0.5000 | 0.0018 | 0.1841 |
| CA | 0.0088 | 0.1517 | 0.0003 | 0.5000 | 0.0017 | 0.0190 | 0.0018 | 0.5000 | 0.1950 |

L = laryngeal, P = pulmonary, CE = cardiac efferent, CA = cardiac afferent, SD = standard deviation.

the parasympathetic efferent pathway via the left vagus nerve. There was no correlation found, during these studies, between the stimulation parameters and the reflex pathway, of the left vagus nerve or sympathetic chain achieving bradycardia and tachycardia, respectively, that was activated by reflex afferent stimulation in each animal. This needs to be investigated further. As seen in Fig. 4*D*, there was a delay in response upon afferent stimulation, following right vagotomy, which could be due to the time required for reflex action via the brainstem or higher centres of the brain, and the subsequent activation of an efferent pathway. For this reason, up to 1 min was incorporated for the off periods between stimulations to allow for this delayed response and corresponding HR changes to take effect and for return to baseline activity before stimulating the sequential pairs. As this extra time was required, in conjunction with testing multiple parameters and this occurring at the end of the allotted experimental time, only every second pair of electrodes was used for this afferent cardiac testing

following vagotomy. Due to the location of stimulation in the cervical neck region, it is predicted that there would be few sympathetic fibres; however, the possibility of affecting vagal sympathetic efferent function via afferent activation cannot be dismissed (Ardell et al., 2015).

Development of the sVNS technique could help address the complexity of performing effective neuromodulation of cardiac function, by targeting individual vagus branches or areas of the nerve responsible for cardiac control; however, a deeper understanding of the functional and

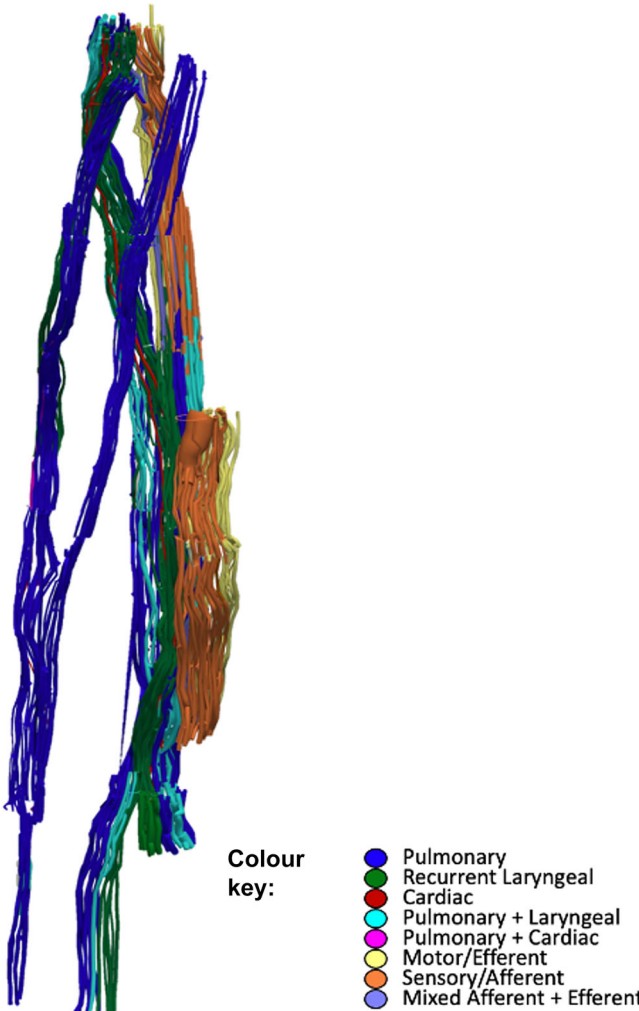

**Colour key:**
- 🔵 Pulmonary
- 🟢 Recurrent Laryngeal
- 🔴 Cardiac
- 🔵 Pulmonary + Laryngeal
- 🟣 Pulmonary + Cardiac
- 🟡 Motor/Efferent
- 🟠 Sensory/Afferent
- 🔵 Mixed Afferent + Efferent

**Figure 6. An example (*n* = 1/5) of 3D segmentation of the fascicles throughout the length of the vagus nerve**
To optimize the microCT scanning of the whole length of the nerve from above the nodose ganglion to below the pulmonary branches, the nerve is cut into segments. These are bundled together within the scanner and so not all segments can be clearly visualized in this 2D example. See colour key for fascicle type identification. The organ-specific (pulmonary, recurrent laryngeal and cardiac) fascicles were traced from the branches exiting the vagus up to the cervical level. The afferent- and efferent-containing fascicles were traced from the nodose ganglion, and those bypassing the ganglion, down to the cervical level. [Colour figure can be viewed at wileyonlinelibrary.com]

anatomical organization of the various cardiac nerve fibres in the cervical vagus is required for this to be possible. Here, we showed that the cardiac efferent and afferent fibres are located separately in the cross-section of the vagus nerve at mid-cervical level, which is a significant advancement in the development of the sVNS approach to modulate cardiac function.

Thompson et al. (2023) found that the fascicles of the superior and inferior cardiac branches merged proceeding up the nerve but maintained separation from the rest of the vagal fascicles at the level of vagal cuff placement at mid-cervical level. This is consistent with what was seen here in these nerves. These fascicles were identified as being efferent/motor by segmentation and tracing from microCT scans of the fascicles from those that bypassed the nodose ganglion. Notably, a significant observation of this study is that the fascicles originating from the superior and inferior cardiac branches from all the nerves investigated were 100% efferent and 100% cardiac, with no merger with any other fascicle types observed. This could be utilized to develop targeted strategies in line with the rationale of this study.

The majority of the fascicles containing either pulmonary fibres or pulmonary and cardiac fibres were identified as afferent/sensory with the fascicles originating in the nodose ganglion. The regions within the cross-section of the nerve containing cardiac efferent and cardiopulmonary afferent fascicles were roughly on opposite sides of the nerve and this corresponded to the activity identified as efferent and afferent cardiac function, respectively, by stimulation of the nerve before and following vagotomy *in vivo*. For the most part, the organization and intraneural distribution of fascicle types across the nerves was quite similar (Tables 1 and 2; Fig. 5). One nerve contained cardiopulmonary fascicles that were mixed as opposed to afferent like the rest of the nerves, and the number and placement of laryngopulmonary fascicles varied between animals.

It must be acknowledged that at the mid-cervical level, there were no unlabelled fascicles due to merger of the traced fascicles with others in the nerve which were subsequently labelled as that organ fibre-containing, and as only thoracic organs were investigated in this study, subdiaphragmatic fibres must be merged with the labelled fascicles at this level. It is hypothesized that subdiaphragmatic fibres, which are known to be mostly afferent, would similarly follow the bimodal distribution observed in the nerve and thereby be located on the afferent, cardiopulmonary and pulmonary side of the nerve.

The organization of vagal fibres related to cardiac function was also investigated in a recent study in mouse (Devarajan et al., 2022), which suggested a convergence of sensory neural pathways of cardiac and respiratory neurons involving a significant fraction

of both (>30%). This is congruent with the finding of cardiac afferents overlapping with the fascicles identified *ex vivo* as containing fibres from pulmonary branches as well as those areas within the nerve identified *in vivo* as having pulmonary activity and cardiac activity following vagotomy (and therefore no efferent activity within the nerve). Furthermore, a recent study molecularly defined two separate cardiac circuits within mice using retrograde tracing, single-cell RNA tracing and optogenetics (Veerakumar et al., 2022). These consisted of the ambiguous cardiovascular circuit which selectively innervates a subset of cardiac parasympathetic ganglion neurons and mediates the baroreceptor reflex, slowing HR and atrioventricular node conduction in response to increased blood pressure; and the other, the ambiguous cardiopulmonary circuit with neurons that intermingled with both the cardiovascular and also innervate most or all lung parasympathetic ganglion neurons thereby having cardiopulmonary function by innervating both organs.

These findings suggest it should be possible to selectively activate the cardiac efferent fibres of the vagus nerve whilst avoiding the cardiac afferents. However, conforming with the aforementioned bimodal distribution of the efferent and afferent fascicles, it is observed that the cardiac efferent fascicles are located

near the recurrent laryngeal fascicles in all nerves in this study. Therefore, it may not be possible to avoid all laryngeal off-target effects when selectively activating the cardiac efferents as some fascicles in the vicinity may be simultaneously activated. Additionally, due to the activation threshold of cardiac fibres being 100 times greater than that for laryngeal fibres, it would be difficult to not activate recurrent laryngeal fibres simultaneously. However, with the target of treating cardiac disease, the avoidance of cardiac afferent fibres that neutralize the activation of efferents is prioritized over the avoidance of minor off-target effects such as cough.

Other efforts in decoding functional signals such as cardiac- or pulmonary-related signals involve the use of invasive methods such as intraneural electrodes in pigs (Vallone et al., 2021) or microelectrodes in awake humans (Patros et al., 2022). Machine learning has also been proposed as a potentially helpful tool for this decoding effort (Pollina et al., 2022). Additionally, optogenetic stimulation of efferents at the level of the dorsal motor nucleus of the vagus nerve has been performed in sheep (Booth et al., 2021).

Recent studies pioneered the concept of a neural fulcrum, based on the notion that when neuromodulation alters autonomic control, the effect is counteracted by

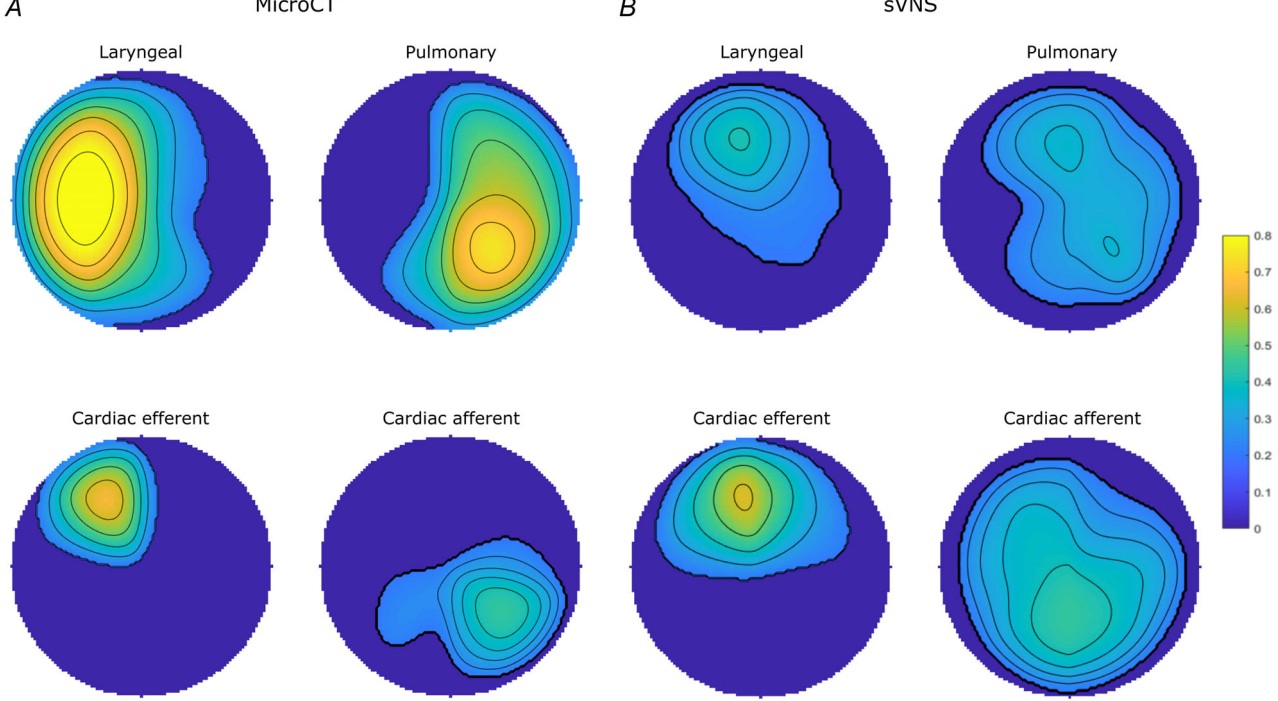

**Figure 7. The mean regions comprising laryngeal, pulmonary, cardiac efferent and cardiac afferent fascicles or function from (*A*) microCT (*n* = 5) and (*B*) electrophysiological (EPhys) data following selective stimulation (*n* = 10), respectively, conformed to a circle for cross-validation**
The colour scale, from 0 to 1, depicts the proportion of fascicles or electrorheological response upon selective activation that were located within the area in the cross-section across all the nerves (0 indicates no fascicles were present at the location in any nerve, 1 indicates that all nerves had a specific fascicle in that location). [Colour figure can be viewed at wileyonlinelibrary.com]

endogenous reflexes (Ardell et al., 2015, 2017). These studies also suggested the use of titration as a way to condition autonomic neural networks over time and implement stimulation protocols with reduced side effects (Ardell et al., 2017). The ANTHEM-HF clinical study, based on the concept of a neural fulcrum and titration, investigated the safety and feasibility of performing cervical VNS in patients with chronic HFrEF (Sharma et al., 2021). In a 42 month follow-up, this therapy was found to be safe and associated with beneficial effects on left ventricular ejection fraction. Titration, however, takes ∼4 weeks to achieve. By selectively stimulating the cardiac efferents and avoiding the majority of the afferent fibres of the vagus nerve, titration could potentially be mitigated for the treatment of cardiac diseases. As shown in Hadaya et al. (2023), considerable cardioprotection was achieved when VNS was applied at the neural fulcrum starting 2 days following MI. However, titration subsequent to MI was required to reach this optimal point. Furthermore, in Ardell et al. (2015), when the vagus nerve was cut centrally, the threshold for bradycardia was markedly reduced. Therefore, using an sVNS approach and activating cardiac efferent fibres whilst avoiding afferents, and thereby not having the concurrent activation of cardiopulmonary,

laryngeal and subdiaphragmatic afferents, therapeutic levels in pure reactive mode could potentially be achieved within days and not weeks. This could save valuable time and improve outcomes and survivability drastically. Future studies in awake unanaesthetized animals with sVNS are required to confirm this conjecture. This selective neuromodulation approach holds potential for extending its therapeutic benefits to atrial fibrillation. Previous research utilizing VNS in a neutrally induced model of atrial fibrillation has highlighted its efficacy in targeting specific populations of intrinsic cardiac neurons within the atria, thereby mitigating neural network imbalances and conferring electrical stability (Salavatian et al., 2016). Considering the demonstrated effectiveness of VNS in this regard, the introduction of a selective approach could further optimize its therapeutic impact on atrial fibrillation, presenting an avenue for enhanced treatment strategies in this condition too. This study, however, did not investigate subdiaphragmatic organs but due to the hypothesized location of GI fibres within the afferent region of the nerve, as well as the exclusive nature of the target fascicles, it is predicted to not have too much of an effect when using a multi-electrode, selective nerve cuff for the purpose of treating cardiac disease and progression to

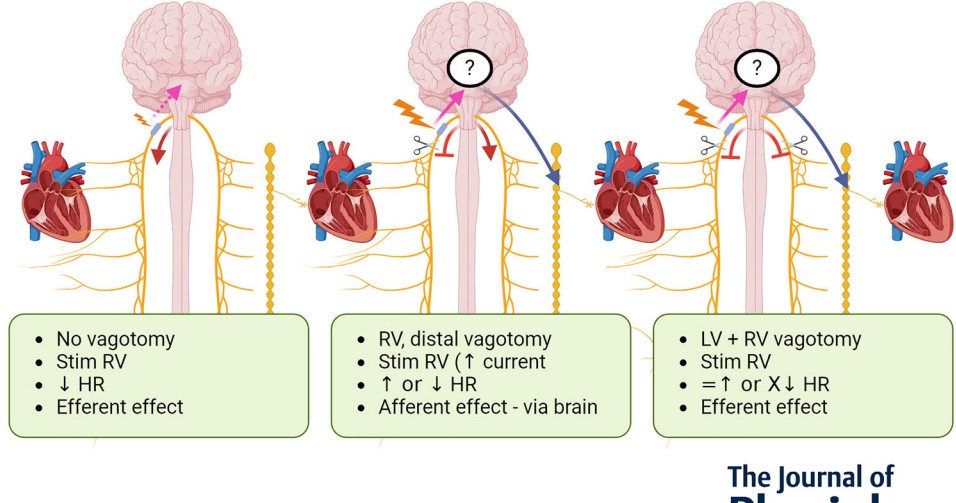

**Figure 8. Effects of vagal stimulation on the heart at different stages in the experiments**
Left: stimulation of the right vagus nerve before vagotomy, achieving efferent effects (red arrow) on the heart from the right vagus nerve. Potential afferent activation (dashed pink arrow) could be possible; however, the stimulation parameters used are at a threshold lower than required to activate the afferent/sensory/smaller fibres but sufficient to predominantly activate the efferent/motor/larger fibres. See Supplementary Fig. 2 for a control performed: the right vagus nerve was activated before and after right proximal vagotomy with little change ($P < 0.05$) in the bradycardia response. Middle: stimulation of the right vagus nerve after right distal vagotomy, eliminating any efferent effects from the right vagus nerve (red stopper) but achieving afferent (pink arrow) effects that reflexively result in either parasympathetic efferent cardiac effects via the left vagus nerve [bradycardia, decrease in heart rate (HR)] (red arrow) or sympathetic cardiac effects via the sympathetic chain (tachycardiac, increase in HR) (blue arrow). Right: stimulation of the right vagus nerve after right distal vagotomy and left vagotomy, resulting in the elimination of efferent effects via either vagus (two red stoppers) which eliminated bradycardia (red stopper) or the persistence of tachycardia (blue arrow) in those animals for which bradycardia or tachycardia was the afferent effect achieved, respectively. [Colour figure can be viewed at wileyonlinelibrary.com]

heart failure following MI. This needs to be investigated further. Titration may still be required, especially for diseases affecting other organs innervated by the vagus.

It was not the aim of the study to decipher the amount of HR drop that would be sufficient for cardioprotection, but rather that we can selectively do so. It was preferential to alter parameters to improve selectivity of HR decrease on the least number of electrodes of the nerve cuff than to alter parameters to get a greater HR decrease on any electrodes. This would be the next steps in MI and/or heart failure models where the parameters and dosing will be adjusted, using the selective, multi-electrode cuff, to elicit the amount of HR decrease sufficient to provide reduction in ventricular arrythmias and other cardiac symptoms associated with MI and heart disease. Our prior experience with whole nerve stimulation, applied early after MI, indicates that vagal nerve efferent stimulation just sufficient to reduce HR by 1–2% is sufficient to mitigate adverse remodelling of the heart and ANS and to substantially reduce the potential for fatal arrhythmias (Hadaya et al., 2023).

While HR response served as the key biomarker for assessing cardiac effects following neuromodulation in this study, it is worth recognizing that the impact of neuromodulation probably extends to ventricular dynamics as well. Through comprehensive preclinical investigations, which encompassed studies utilizing atrial and ventricular indices alongside an exploration of the structure–function organization of the thoracic vagosympathetic nerve trunk, valuable insights into the multifaceted effects of neuromodulation on cardiac function were gained (Ardell et al., 2015, 2017; Dacey et al., 2022; Randall et al., 1985). Notably, results obtained using observed chronotropic effects are paralleled by dromotropic and inotropic effects in both atrial and ventricular tissues, highlighting the broad-reaching influence of neuromodulation on various aspects of cardiac physiology (Dacey et al., 2022; Randall et al., 1985). Based on the observed HR changes in this study utilizing the novel approach of sVNS, it is anticipated that a similar correlation between control of other heart functions will persist.

This research study, despite its promising findings, has certain inherent limitations that need to be acknowledged. First and foremost, the high amplitudes required for successful sVNS may not be easily translatable to human applications, as the threshold for stimulation in humans might differ significantly from the observed levels. Despite pulmonary testing being performed when the ventilation was paused and the animal was spontaneously breathing, it is difficult to ascertain the effects sVNS had on the lungs during ventilation when testing other organs. Additionally, despite bilateral vagotomy at the end of the experiment to control for left efferent input, the study did not address the potential influence of other neural circuit pathways that were left intact, such as the sympathetic chain and other spinal cord inputs. Understanding the interplay between these pathways and the effects of their preservation on the outcomes could provide a more comprehensive picture. Further elucidation of the mechanisms underlying the inconsistent effects of vagal afferent fibre activation on HR is warranted, particularly regarding the potential reflexive activation of sympathetic efferent pathways or left vagus nerve efferents. Given the complexities involved and the extensive scope of potential pathways and physiological factors to consider, elucidating these mechanisms falls beyond the confines of the current study. This uncertainty not only emphasizes the need for further investigation to better understand the reflex pathways in the brainstem and higher centres of the brain but it also further supports the rationale of selectively targeting efferent fibres. Furthermore, the study utilized pig models, and the translatability of results to humans may be limited due to differences in fascicle number, size and distribution between the two species. Finally, the potential contribution of co-release of neuromediators (e.g. peptides) during sVNS should be considered in addition to the primary postganglionic neurotransmitters. These limitations underscore the need for further investigation and refinement before the full clinical potential of sVNS can be realized.

Future work will encompass additional and chronic animal studies with MI and heart failure models, understanding and improving stimulation parameters and the target HR decrease to achieve cardioprotection, as well as future human trials, as this is essential for validating the potential of sVNS in addressing heart failure, MI and other applications. Our upcoming advancements in technology, including a battery-free, off-the-shelf sVNS implantable stimulator and a pressure-sensitive sVNS cuff to prevent excessive nerve pressure, will contribute significantly to enhancing the feasibility and effectiveness of sVNS for these clinical purposes.

This study found significant spatial separation between cardiac afferent and efferent fibres at the mid-cervical level and, notably, cardiac efferent fascicles were exclusive. Our study demonstrated that targeted neuromodulation via sVNS could achieve scalable HR decreases without eliciting cardiac afferent-related reflexes; this is desirable for reducing sympathetic overactivation associated with heart disease.

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

## Additional information

### Data availability statement

The data from the research in this paper will be made available on Pennsieve with the following DOI: 10.26275/0bdd-rwbe.

### Competing interests

The authors declare that the research was conducted in the absence of any commercial or financial relationships that could be construed as a potential conflict of interest.

### Author contributions

N.T., E.R., J.A., D.H. and K.A. conceived and designed the research; N.T., E.R., S.M., R.C., J.H. and K.A. performed

the experiments; N.T., E.R. and K.A. analysed the data and interpreted the results of the experiment; N.T. prepared the figures and tables; F.I., A.S.I. and P.S. contributed resources and equipment to the study; O.A., J.A., K.S. and D.H. provided supervision, funding acquisition, administration and guidance; N.T. and E.R. drafted the manuscript; N.T., S.M., O.A., J.A., K.S., D.H. and K.A. edited and revised the manuscript. All authors approved the final version of the manuscript; agree to be accountable for all aspects of the work in ensuring that questions related to the accuracy or integrity of any part of the work are appropriately investigated and resolved; and all persons designated as authors qualify for authorship, and all those who qualify for authorship are listed.

## Funding

This work was supported by EPSRC EP/X018415/1 (N.T., E.R., A.S.I., D.H., K.A.), NIH SPARC 1OT2OD026545-01 (N.T., E.R., S.M., A.S.I., D.H., K.A.), RO1 HL162921-01A1 (J.L.A., K.S.), NIH SPARC 1OT2OD023848 (J.L.A., K.S., O.A.), and P01 HL164311-01A1 (J.L.A., K.S., O.A.).

## Keywords

afferent and efferent, cardiac, heart failure, neuromodulation, spatially selective, vagus nerve stimulation (VNS)

## Supporting information

Additional supporting information can be found online in the Supporting Information section at the end of the HTML view of the article. Supporting information files available:

**Peer Review History**
**Supplementary Information**

