## [Peer Review History · The Journal of Physiology]

Towards spatially selective efferent neuromodulation: Anatomical and functional organization of cardiac fibers in the porcine cervical vagus nerve

Nicole Thompson, Enrico Ravagli, Svetlana Mastitskaya, Ronald Challita, Joseph Hadaya, Francesco Iacoviello, Ahmad Shah Idil, Paul R Shearing, Olujimi A Ajjola, Jeffrey Laurence Ardell, Kalyanam Shivkumar, David Holder, and Kirill Aristovich

DOI: 10.1113/JP286494

Corresponding author(s): Nicole Thompson (nicole.thompson@ucl.ac.uk)

The following individual(s) involved in review of this submission have agreed to reveal their identity: Fabio Recchia (Referee #2)

Review Timeline:

Submission Date:	01-Mar-2024
Editorial Decision:	07-May-2024
Revision Received:	28-Jun-2024
Editorial Decision:	23-Jul-2024
Revision Received:	06-Aug-2024
Editorial Decision:	07-Aug-2024
Revision Received:	07-Aug-2024
Accepted:	09-Aug-2024

Senior Editor: Harold Schultz

Reviewing Editor: Andrew Holmes

Transaction Report:

Dear Dr Thompson,

Re: JP-RP-2024-286494 "Anatomical and functional organization of cardiac fibers in the porcine cervical vagus nerve allows spatially selective efferent neuromodulation" by Nicole Thompson, Enrico Ravagli, Svetlana Mastitskaya, Ronald Challita, Joseph Hadaya, Francesco Iacoviello, Ahmad Shah Idil, Paul R Shearing, Olujimi A Ajjola, Jeffrey Laurence Ardell, Kalyanam Shivkumar, David Holder, and Kirill Aristovich

Thank you for submitting your manuscript to The Journal of Physiology. It has been assessed by a Reviewing Editor and by 2 expert referees and we are pleased to tell you that it is potentially acceptable for publication following satisfactory major revision.

LANGUAGE EDITING AND SUPPORT FOR PUBLICATION: If you would like help with English language editing, or other article preparation support, Wiley Editing Services offers expert help, including English Language Editing, as well as translation, manuscript formatting, and figure formatting at www.wileyauthors.com/eoo/preparation. You can also find resources for Preparing Your Article for general guidance about writing and preparing your manuscript at www.wileyauthors.com/eoo/prepresources.

REVISION CHECKLIST:

Please upload two versions of your manuscript text: one with all relevant changes highlighted and one clean version with no changes tracked. The manuscript file should include all tables and figure legends, but each figure/graph should be uploaded as separate, high-resolution files. The journal is now integrated with Wiley's Image Checking service. For further details,

see: <https://www.wiley.com/en-us/network/publishing/research-publishing/trending-stories/upholding-image-integrity-wileys-image-screening-service>

We look forward to receiving your revised submission.

Yours sincerely,

Harold Schultz
Senior Editor
The Journal of Physiology

REQUIRED ITEMS FOR REVISION

- Author photo and profile. First or joint first authors are asked to provide a short biography (no more than 100 words for one author or 150 words in total for joint first authors) and a portrait photograph. These should be uploaded and clearly labelled together in a Word document with the revised version of the manuscript. See Information for Authors for further details.

- You must start the Methods section with a paragraph headed Ethical Approval. A detailed explanation of journal policy and regulations on animal experimentation is given in Principles and standards for reporting animal experiments in The Journal of Physiology and Experimental Physiology by David Grundy J Physiol, 593: 2547-2549. doi:10.1113/JP270818). A checklist outlining these requirements and detailing the information that must be provided in the paper can be found at: <https://physoc.onlinelibrary.wiley.com/hub/animal-experiments>. Authors should confirm in their Methods section that their experiments were carried out according to the guidelines laid down by their institution's animal welfare committee, and conform to the principles and regulations as described in the Editorial by Grundy (2015), including an ethics approval reference number. The Methods section must contain a statement about access to food, water and housing, details of the anaesthetic regime: anaesthetic used, dose and route of administration, and method of killing the experimental animals.

- The reference list must be in alphabetical order, rather than numbered, to comply with our Journal format.

- Your manuscript must include a complete Additional Information section, including competing interests; funding; author contributions and acknowledgements.

- Please upload separate high-quality figure files via the submission form.

- Please ensure that any tables are editable and in Word format, and wherever possible, embedded in the article file itself.

- Papers must comply with the Statistics Policy: https://jp.msubmit.net/cgi-bin/main.plex?form_type=display_requirements#statistics.

In summary:

- If n {less than or equal to} 30, all data points must be plotted in the figure in a way that reveals their range and distribution. A bar graph with data points overlaid, a box and whisker plot or a violin plot (preferably with data points included) are acceptable formats.

- If $n > 30$, then the entire raw dataset must be made available either as supporting information, or hosted on a not-for-profit repository, e.g. FigShare, with access details provided in the manuscript.
- 'n' clearly defined (e.g. x cells from y slices in z animals) in the Methods. Authors should be mindful of pseudoreplication.
- All relevant 'n' values must be clearly stated in the main text, figures and tables.
- The most appropriate summary statistic (e.g. mean or median and standard deviation) must be used. Standard Error of the Mean (SEM) alone is not permitted.
- Exact p values must be stated. Authors must not use 'greater than' or 'less than'. Exact p values must be stated to three significant figures even when 'no statistical significance' is claimed.

- Please include an Abstract Figure file, as well as the Figure Legend text within the main article file. The Abstract Figure is a piece of artwork designed to give readers an immediate understanding of the research and should summarise the main conclusions. If possible, the image should be easily 'readable' from left to right or top to bottom. It should show the physiological relevance of the manuscript so readers can assess the importance and content of its findings. Abstract Figures should not merely recapitulate other figures in the manuscript. Please try to keep the diagram as simple as possible and without superfluous information that may distract from the main conclusion(s). Abstract Figures must be provided by authors no later than the revised manuscript stage and should be uploaded as a separate file during online submission labelled as File Type 'Abstract Figure'. Please also ensure that you include the figure legend in the main article file. All Abstract Figures should be created using BioRender. Authors should use The Journal's premium BioRender account to export high-resolution images. Details on how to use and access the premium account are included as part of this email.

EDITOR COMMENTS

Reviewing Editor:

Thank you for submitting the manuscript to The Journal of Physiology. We appreciate the time and effort that goes into such submissions. The work is interesting and timely, and experiments are performed well. That said there are some very important points raised by the reviewers that need to be addressed. Furthermore, the manuscript should be extensively modified to make it easier for the reader to identify the key points and messages. This could even start with the title conveying a slightly clearer message. I agree with the reviewers that some more analysis/presentation of functional data (including statistics) would help to emphasise the importance of the anatomical observations. I think there should also be more discussion of the work in the context of what is already known, and the reviewers point out some useful additional references. In addition please can the following information be provided:

1. Please can the source of the animals be stated in the methods
2. Please can you state whether or not animals had free access to food and water
3. Please can you provide a graphical abstract that clearly highlights the main findings of the work.

Senior Editor:

Comments for Authors to ensure the paper complies with the Statistics Policy:
Variance is not defined for means. The Journal requires variance as standard deviation or box whisker plot quartiles.

Statistical analysis must be described in the methods in a section at the end titled Statistical Analysis.

Comments to the Author:

Thank you for submission of your research article to the Journal of Physiology for consideration. The article has been reviewed by experts in the field and found to require extensive revision before a decision on publication can be reached. The revision must address all the concerns raised by referees and editors. Please also address the list of requirements or publication in the journal including manuscript formatting, animal ethics, statistical requirements, and supplementary

material.

1. General: The manuscript needs to be rewritten to more succinctly focus on the important novel findings. Many places are not clearly written, are repetitive, off topic or overreaching interpretation of the data presented. These issues cloud the ability to follow the logic of a complex study design and results to reveal clear conclusions. Both referees state, and editors agree, that the conclusions of the study are too far reaching, and sometimes go against the data presented. It is deemed inclusive that the spatially selected stimulation targets cardiac efferent fibers only, particularly regarding vagal nerve function to organs other than the heart and lungs not assessed (e.g. GI which possesses a large proportion of vagal fibers). In this regard, the title, abstract and many parts in the manuscript overreach the conclusions for selective efferent neuromodulation. It is recommended to concentrate on the identification of a spatial separation of the cardiac afferent and cardiac efferent regions as the important end point. Please revise these parts of the manuscript to reflect this.

2. Please review the journal policy on formatting of a research manuscript: https://jp.msubmit.net/cgi-bin/main.plex?form_type=display_requirements#Revised%20submissions

2a. The abstract must be written to journal policy (see above link). The Journal does not accept a condensed abstract as a substitute for an inadequate full abstract. As written, the background is nearly half of the abstract. This should be reduced to one or two sentences. The purpose or aim of the study must be succinctly and clearly stated. Provide brief description of the research methods and study design. State key techniques. A major part of the abstract should clearly identify the results of the experiments, expanding and supporting the conclusions drawn in the abstract.

2b. Please review the journal policy on reporting animal ethics: <https://physoc.onlinelibrary.wiley.com/hub/animal-experiments>

The Methods section must begin with the subheading "Ethical Approval" wherein authors must state the institutional ethical committee that approved the study and the national guidelines under which the institution operates. The institutional approval code/number should be provided. In addition, the section must include a statement that the investigators understand the ethical principles under which the journal operates and that their work complies with the animal ethics checklist as outlined by the journal.

Identify the source. Detail the supplier or commercial breeder. Specify species, strain, genetic background, weight, sex and age. Indicate the group size and total numbers of animals, including any animals used but subsequently excluded for any reason, specifying any unexpected events. For farm animals it is not necessary to name the commercial supplier. However, information on the country's regulatory authority governing animal husbandry and transportation must be provided. For example: "All animals were reared and transported under conditions specified in the UK's Animal Welfare Act 2006 and The Welfare of Farm Animals (England) Regulations 2007."

State the feeding and water regime e.g. fasting, feeding and water ad libitum or on a specific diet.

For in vivo experiments under acute surgical anesthesia, please provide an estimate of the total duration of the experiment from induction of anesthesia to endpoint of measurements. An experimental timeline would be very helpful.

Although the manuscript covers the anesthesia protocol in sufficient detail and how the depth of anesthesia was determined and maintained through the experiment, it was not clear when the type of anesthesia was switched from isoflurane to alpha-chloralose. Line 240 states after surgical procedures, but line 271 states "After the initial round of sVNS for identification of pulmonary fascicles, the animal was put back on mechanical ventilation and anesthesia was switched to α -chloralose...". When were the animals switched back to isoflurane, when were they taken off mechanical ventilation? It appears that these statements are conflicting. It would help with clarity if the surgical preparation and experimental procedures were separate sections. Again, an experimental timeline would greatly help.

3. Please review journal policy on statistical reporting: https://jp.msubmit.net/cgi-bin/main.plex?form_type=display_requirements#statistics

The methods section must conclude with a section on data and statistical analysis. It is not clear how variance for means was measured. The Journal requires variance as standard deviation or box whisker plot quartiles. For a given conclusion to be assessed, the exact p values must be stated to three significant figures, unless $p < 0.001$, where $p < 0.001$ is acceptable. Please state samples size and x samples in x animals in text and figure/table legends. Indicate the statistical test used in figures and tables. If uniform throughout, state in the statistical analysis section of methods or how each test was applied to individual experiments.

4. Supplementary material: The citation of supplementary sheets in the manuscript is very confusing. The supplement obtained is an Excel file that does not identify material as Sheets 1-5. The file is a combination of tables, figures and data with titles but not in any order to identify sheets. And supplement Fig 1 is missing. There is one sheet with just a key. As per Journal policy the figures need to be incorporated into the manuscript. This rule applies also to tables that are needed to interpret results for conclusions. Any remaining supplementary material must be submitted as a PDF file(s). The Excel file also had links to external source(s), that may be blocked by institutional IT systems that prohibit embedded links to external sources (indeed an issue for this editor). Any link to external data must be cited with a full URL to the source, allowing access outside of institutional firewalls.

REFeree COMMENTS

Referee #2:

The authors used trial-and-error selective vagal nerve stimulation (sVNS) in vivo in combination with ex vivo micro-computed tomography fascicle tracing and showed a significant spatial separation of cardiac afferent and efferent fibers (179 ± 55 degree sign SD microCT, $p < 0.05$ and 200 ± 137 degree sign SD, $p < 0.05$ sVNS - degrees of separation across a cross-section of nerve) at the mid-cervical level. They also showed that cardiac afferent fibers are located in proximity to pulmonary fibers, consistent with recent findings of cardiopulmonary convergent neurons and circuits. They demonstrated the ability of sVNS to selectively elicit desired scalable heart rate decrease without stimulating afferent-related reflexes. They concluded that their findings pave the way for more targeted neuromodulation, thereby reducing off-target effects and eliminating the need for titration.

This study was performed by recognized experts in the field of vagus nerve physiology who utilized state of the art methodologies. Its main limitation is the moderate novelty relative to other studies.

Major comments

1. The authors should refocus their manuscript on the really novel findings (and should adequately highlight them), if any, relative to other studies that have also aimed at defining the spatial distribution and the functional role of cardiac efferent vs afferent vagal nerve fibers. For instance, the recent study by Jayaprakash et al (ref. 21 in the manuscript) has provided a massive amount of anatomical data on efferent vs afferent topography by employing microCT and immunohistochemistry as well.

2. Both in the Introduction and in the Discussion, the authors put much emphasis on the need highly selective VNS for to induce cardioprotection while mitigating off-target effects by specifically targeting pre-ganglionic parasympathetic efferent cardiac fibers. Moreover, they claim a demonstrated ability of sVNS "to selectively elicit desired scalable heart rate decrease". However, the results of functional tests are presented in a single, succinct paragraph on page 18, where the authors report relatively small changes in heart rate without indicating any statistical significance. In addition, there are no data showing selective elicitation of desired scalable heart rate decrease. Once again, what is the difference relative to previous studies, some of which tested vagal stimulations even in chronically implanted animals? This section of the study needs to be expanded.

3. Figure 4: it seems these sections were taken at the same level in 5 different animals. The intraneural distribution of the identified fiber types varies notably from animal to animal. Why?

Minor comments

1. In Abstract, Introduction and Discussion, the authors write at length about the clinical benefits of sVNS for the treatment of cardiac diseases. These parts of the text can be shortened, as the present study did not test any of such therapeutic interventions. The authors' considerations should instead be more focused on the actual findings and their relevance relative to previous studies.

2. Some parts of the manuscript are repetitive: for instance, lines 173-175 in Introduction and lines 240-242 in Methods, but also others.

Referee #3:

Autonomic imbalance is a significant indicator of adverse clinical outcomes in many cardiovascular diseases. Vagal nerve stimulation has emerged as a popular treatment option for CV diseases. However, the current cervical vagal stimulation devices have off-target side effects due to the lack of cardiac specificity. To address this, Thompson et al. investigated the spatial organisation of the cervical vagus. They tested the feasibility of selective vagal nerve stimulation (sVNS) using a multi-electrode cuff the authors have previously developed.

Thompson and colleagues conducted a study using an anaesthetised pig model, combining functional and paired anatomical investigations. The study's significant strength is the paired data approach. Anatomical micro-CT studies revealed the spatial separation of cardiac afferent and efferent fibres and some spatial specificity of laryngeal and pulmonary fibres. Functional studies demonstrate the feasibility of spatially selective heart rate control; however, cross-activation of other fibres does occur. The study is well designed, given the constraints of achieving such technically challenging experiments. The methods are extensively detailed and very sound; however, some aspects would benefit from clarification. While the study presents compelling data on the potential feasibility of spatially selected vagal nerve stimulation, some conclusions go beyond the data presented and would benefit from additional information.

Major comments:

- The authors present elegant data showing the spatial anatomical location of cardiac, pulmonary and laryngeal-containing fascicles in the cervical vagus but provide no data on the other fibre types present. It would be near impossible to micro-CT the whole vagus. However, providing additional data on the number of efferent and afferent fascicles that are not originating in the neck/thorax, i.e. any fibres that can be traced to the nodose but not to the heart, lungs, or larynx, would give valuable information on just how specific cardiac efferent stimulation could be.

- In line with the above, stating that targeted cardiac efferent stimulation can be achieved 'without stimulating afferent reflexes' (Line 52, and others), should say 'cardiac-afferent reflexes'. Other non-cardiac afferent reflexes that are not being recorded from may be triggered. Moreover, it is difficult to ascertain how sVNS may influence the lungs when the animal is ventilated.

- The authors focus on how the data presented will 'eliminate the need for titration' and the benefits of achieving faster onset of therapeutic VNS. While this may be a potential benefit of target stimulation, there is no data to support this and stating this as fact is beyond the scope of the study. The authors present data showing that targeted cardiac efferent stimulation will always activate laryngeal fibres - as cough and GI issues are the major side effects that VNS titration aims to avoid the data presented in its current form does not suggest that the need for titration would be 'eliminated'.

Minor comments:

Introduction:

- Citing some of the original cardiac VNS literature (e.g. Sunagawa's Circulation paper from 2003) would be beneficial, instead of just citing articles from the research team.
- Introduction is, at times, repetitive and could be shortened.

Methods:

- How many pigs were done in each location? Was there any failure rate in the functional studies?
- Clarifying how many studies were paired, n = 5? and how many were functional-only studies would help transparency.
- The experimental timeline paragraph (lines 210-221) could be better presented as a timeline.
- Why were there stimulation electrodes on the recurrent laryngeal nerve?
- When was fentanyl stopped?
- Was the left vagus also stimulated or recorded during the procedure (line 258)? Direct recording from the left vagus after right vagal afferent stimulation would be interesting.
- Were the pulmonary responses tested under isoflurane anaesthesia? How many Ephys animals underwent pulmonary testing? This appears low in the supplemental data (thank you for the extensive supplemental data)- The addition of specific n to figures would help transparency.

Results:

- With data presented as per cent change, please provide a table of raw data values.
- Figure 3. A great raw data figure, but some aspects do not match what is written in the methods. On/off periods are stated to be 15 seconds for everything but laryngeal stimulation, but panel D shows something different. Heart rates are lower than what is stated as the target range in the methods. Why are some electrode combinations missing from the afferent stimulation?

Discussion

- Rationale for using the right VNS.
- Line 469: The number of cardiac efferent fascicles is low, but if they are 100% efferent when originating from the cardiac vagal branch, this is a significant observation! And could help develop targeted VNS strategies.

END OF COMMENTS

Confidential Review

01-Mar-2024

Autonomic imbalance is a significant indicator of adverse clinical outcomes in many cardiovascular diseases. Vagal nerve stimulation has emerged as a popular treatment option for CV diseases. However, the current cervical vagal stimulation devices have off-target side effects due to the lack of cardiac specificity. To address this, Thompson et al. investigated the spatial organisation of the cervical vagus. They tested the feasibility of selective vagal nerve stimulation (sVNS) using a multi-electrode cuff the authors have previously developed.

Thompson and colleagues conducted a study using an anaesthetised pig model, combining functional and paired anatomical investigations. The study's significant strength is the paired data approach. Anatomical micro-CT studies revealed the spatial separation of cardiac afferent and efferent fibres and some spatial specificity of laryngeal and pulmonary fibres. Functional studies demonstrate the feasibility of spatially selective heart rate control; however, cross-activation of other fibres does occur. The study is well designed, given the constraints of achieving such technically challenging experiments. The methods are extensively detailed and very sound; however, some aspects would benefit from clarification. While the study presents compelling data on the potential feasibility of spatially selected vagal nerve stimulation, some conclusions go beyond the data presented and would benefit from additional information.

Major comments:

- The authors present elegant data showing the spatial anatomical location of cardiac, pulmonary and laryngeal-containing fascicles in the cervical vagus but provide no data on the other fibre types present. It would be near impossible to micro-CT the whole vagus. However, providing additional data on the number of efferent and afferent fascicles that are not originating in the neck/thorax, i.e. any fibres that can be traced to the nodose but not to the heart, lungs, or larynx, would give valuable information on just how specific cardiac efferent stimulation could be.
- In line with the above, stating that targeted cardiac efferent stimulation can be achieved 'without stimulating afferent reflexes' (Line 52, and others), should say 'cardiac-afferent reflexes'. Other non-cardiac afferent reflexes that are not being recorded from may be triggered. Moreover, it is difficult to ascertain how sVNS may influence the lungs when the animal is ventilated.
- The authors focus on how the data presented will 'eliminate the need for titration' and the benefits of achieving faster onset of therapeutic VNS. While this may be a potential benefit of target stimulation, there is no data to support this and stating this as fact is beyond the scope of the study. The authors present data showing that targeted cardiac efferent stimulation will always activate laryngeal fibres – as cough and GI issues are the major side effects that VNS titration aims to avoid the data presented in its current form does not suggest that the need for titration would be 'eliminated'.

Minor comments:

Introduction:

- Citing some of the original cardiac VNS literature (e.g. Sunagawa's Circulation paper from 2003) would be beneficial, instead of just citing articles from the research team.

- Introduction is, at times, repetitive and could be shortened.

Methods:

- How many pigs were done in each location? Was there any failure rate in the functional studies?
- Clarifying how many studies were paired, $n = 5$? and how many were functional-only studies would help transparency.
- The experimental timeline paragraph (lines 210-221) could be better presented as a timeline.
- Why were there stimulation electrodes on the recurrent laryngeal nerve?
- When was fentanyl stopped?
- Was the left vagus also stimulated or recorded during the procedure (line 258)? Direct recording from the left vagus after right vagal afferent stimulation would be interesting.
- Were the pulmonary responses tested under isoflurane anaesthesia? How many Ephys animals underwent pulmonary testing? This appears low in the supplemental data (thank you for the extensive supplemental data)– The addition of specific n to figures would help transparency.

Results:

- With data presented as per cent change, please provide a table of raw data values.
- Figure 3. A great raw data figure, but some aspects do not match what is written in the methods. On/off periods are stated to be 15 seconds for everything but laryngeal stimulation, but panel D shows something different. Heart rates are lower than what is stated as the target range in the methods. Why are some electrode combinations missing from the afferent stimulation?

Discussion

- Rationale for using the right VNS.
- Line 469: The number of cardiac efferent fascicles is low, but if they are 100% efferent when originating from the cardiac vagal branch, this is a significant observation! And could help develop targeted VNS strategies.

Comments for the Editor

Thompson et.al. present a technically challenging study aimed to establish the functional and morphological spatial selectivity of cardiac vagal nerves within the main (cervical) vagal nerve. This research is highly topical given the clinical interest in developing better vagal stimulation therapies. Some insight is provided into the physiological mechanisms underlying spatially selective efferent neuromodulation given the paired functional and anatomical studies –a strength of this paper. The methods are rigorous and well described, but would benefit from additional clarification around ‘n’ numbers in some parts to improve transparency. Some of the conclusions made go beyond, and sometimes against, the data presented. This is predominantly in relation to how spatially selected stimulation can be used to target cardiac efferent fibres only. The authors themselves state that stimulation of cardiac efferent fibres always co-stimulates laryngeal fibres, and no indication is given for what other fascicles (e.g. GI) may be stimulated with cardiac efferent stimulation. The anatomical data and micro-CT tracing are to be commended and advance our understanding of the mapping of vagal nerves in the pig, building on Thompson’s 2023 paper. This study still has the potential to impact the field and influence how future studies approach vagal nerve stimulation. Specific improvements have been made to the author which would improve the validity of the conclusions.

Recommendation on acceptability

The experiments are methodologically sound, and the data is interesting. Although some conclusions made by the authors go beyond what the data shows, what the study presents has the potential to impact how target vagal stimulation is advanced in the future. Recommend accepting with additional data and revisions.

Comments on influence

This study could have a significant impact on the field of vagal nerve stimulation, helping to advance our understanding of the anatomy and mapping of the vagal nerve. Further insights into how afferent and efferent innervation in the vagus can be selectively targeted are of great clinical interest.

Responses

REQUIRED ITEMS FOR REVISION

- Author photo and profile. First or joint first authors are asked to provide a short biography (no more than 100 words for one author or 150 words in total for joint first authors) and a portrait photograph. These should be uploaded and clearly labelled together in a Word document with the revised version of the manuscript. See Information for Authors for further details.

We have collated this into a Word document and will upload it upon submission.

Nicole Thompson, a post-doctoral fellow at UCL's Neurophysiology and Electrical Impedance Tomography Research Group, holds a PhD in Neuroscience and Biomedical Engineering from UCL. Specializing in neuroimaging and neurophysiology, her research targets imaging techniques, medical devices, and vagus nerve stimulation (VNS), focusing on mammalian vagus nerve anatomy. She aims to enhance understanding of vagus nerve anatomy for improved neuromodulation, particularly in treating epilepsy and heart failure. She is currently coordinating a VNS clinical trial.

Enrico Ravagli is a Senior Research Fellow at UCL's Neurophysiology and Electrical Impedance Tomography Research Group and holds a PhD in Bioengineering from the University of Bologna. His research interests include neuromodulation of the vagus nerve, Electrical Impedance Tomography (EIT) of neural tissue, and developing open-source biomedical instrumentation. He is currently the holder of a UKRI fellowship and a Visiting Research at the University of Sydney, Australia.

- You must start the Methods section with a paragraph headed Ethical Approval. A detailed explanation of journal policy and regulations on animal experimentation is given in Principles and standards for reporting animal experiments in The Journal of Physiology and Experimental Physiology by David Grundy (J Physiol, 593: 2547-2549. doi:10.1113/JP270818). A checklist outlining these requirements and detailing the information that must be provided in the paper can be found at: <https://physoc.onlinelibrary.wiley.com/hub/animal-experiments>. Authors should confirm in their Methods section that their experiments were carried out according to the guidelines laid down by their institution's animal welfare committee, and conform to the principles and regulations as described in the Editorial by Grundy (2015), including an ethics approval reference number. The Methods section must contain a statement about access to food, water and housing, details of the anaesthetic regime: anaesthetic used, dose and route of administration, and method of killing the experimental animals.

We have since added a section title as advised, moved the relevant information from our current methods up to this section, and added the further required information. Please see the highlighted text.

- The reference list must be in alphabetical order, rather than numbered, to comply with our Journal format.

The reference style has been updated to the format of this journal.

- Your manuscript must include a complete Additional Information section, including competing interests; funding; author contributions and acknowledgements.

The missing components (author contributions) has been added to the additional information section at the end of the manuscript.

- Please upload separate high-quality figure files via the submission form.

We will do this upon submission.

- Please ensure that any tables are editable and in Word format, and wherever possible, embedded in the article file itself.

We have updated the table (Table 1) within the manuscript to be in Word format.

- Papers must comply with the Statistics Policy: https://jp.msubmit.net/cgi-bin/main.plex?form_type=display_requirements#statistics.

In summary:

- If n {less than or equal to} 30, all data points must be plotted in the figure in a way that reveals their range and distribution. A bar graph with data points overlaid, a box and whisker plot or a violin plot (preferably with data points included) are acceptable formats.

- If $n > 30$, then the entire raw dataset must be made available either as supporting information, or hosted on a not-for-profit repository, e.g. FigShare, with access details provided in the manuscript.

- 'n' clearly defined (e.g. x cells from y slices in z animals) in the Methods. Authors should be mindful of pseudoreplication.

- All relevant 'n' values must be clearly stated in the main text, figures and tables.

- The most appropriate summary statistic (e.g. mean or median and standard deviation) must be used. Standard Error of the Mean (SEM) alone is not permitted.

- Exact p values must be stated. Authors must not use 'greater than' or 'less than'. Exact p values must be stated to three significant figures even when 'no statistical significance' is claimed.

N numbers were included throughout the paper; however, even more (highlighted within the text) have been added for further clarity.

Majority of the results were reported as a value \pm standard deviation. We have added a line at the beginning of the results to clarify this. Apologies for the confusion. In cases where the standard deviation was not given (end of section 3.2), this has been added.

The p values have been updated within the text to state the exact, 3-decimal point value, as well as shown in a new table from the supplementary information, now incorporated in the manuscript (Table 2).

- Please include an Abstract Figure file, as well as the Figure Legend text within the main article file. The Abstract Figure is a piece of artwork designed to give readers an immediate understanding of the research and should summarise the main conclusions. If possible, the image should be easily 'readable' from left to right or top to bottom. It should show the

physiological relevance of the manuscript so readers can assess the importance and content of its findings. Abstract Figures should not merely recapitulate other figures in the manuscript. Please try to keep the diagram as simple as possible and without superfluous information that may distract from the main conclusion(s). Abstract Figures must be provided by authors no later than the revised manuscript stage and should be uploaded as a separate file during online submission labelled as File Type 'Abstract Figure'. Please also ensure that you include the figure legend in the main article file. All Abstract Figures should be created using BioRender. Authors should use The Journal's premium BioRender account to export high-resolution images. Details on how to use and access the premium account are included as part of this email.

We have created an Abstract figure and included this with a legend after the abstract section of the manuscript.

EDITOR COMMENTS

Reviewing Editor:

Thank you for submitting the manuscript to The Journal of Physiology. We appreciate the time and effort that goes into such submissions. The work is interesting and timely, and experiments are performed well. That said there are some very important points raised by the reviewers that need to be addressed. Furthermore, the manuscript should be extensively modified to make it easier for the reader to identify the key points and messages. This could even start with the title conveying a slightly clearer message. I agree with the reviewers that some more analysis/presentation of functional data (including statistics) would help to emphasise the importance of the anatomical observations. I think there should also be more discussion of the work in the context of what is already known, and the reviewers point out some useful additional references. In addition please can the following information be provided:

Thank you for your comments. We have since modified the manuscript with all the changes suggested by the editors and reviewers and agree that these points needed to be addressed and believe the manuscript is much improved.

1. Please can the source of the animals be stated in the methods

We have since added this information to the manuscript.

2. Please can you state whether or not animals had free access to food and water

We have since added this information to the manuscript.

3. Please can you provide a graphical abstract that clearly highlights the main findings of the work.

We have created an Abstract figure and included this with a legend after the abstract section of the manuscript.

Senior Editor:

Comments for Authors to ensure the paper complies with the Statistics Policy:
Variance is not defined for means. The Journal requires variance as standard deviation or box whisker plot quartiles.

Majority of the results were reported as a value \pm standard deviation. We have added a line at the beginning of the results to clarify this. Apologies for the confusion. In cases where the standard deviation was not and could be given (end of section 3.2), this has been added. The p values have been updated within the text to state the exact, 3-decimal point value, as well as shown in a new table from the supplementary information, now incorporated in the manuscript (Table 2).

Statistical analysis must be described in the methods in a section at the end title Statistical Analysis.

It is currently in a section title "Data co-registration and statistical analyses" – this has been updated to just "Statistical Analysis" in accordance with the editor and journal requirements.

Comments to the Author:

Thank you for submission of your research article to the Journal of Physiology for consideration. The article has been reviewed by experts in the field and found to require extensive revision before a decision on publication can be reached. The revision must address all the concerns raised by referees and editors. Please also address the list of requirements or publication in the journal including manuscript formatting, animal ethics, statistical requirements, and supplementary material.

1. General: The manuscript needs to be rewritten to more succinctly focus on the important novel findings. Many places are not clearly written, are repetitive, off topic or overreaching interpretation of the data presented. These issues cloud the ability to follow the logic of a complex study design and results to reveal clear conclusions. Both referees state, and editors agree, that the conclusions of the study are too far reaching, and sometimes go against the data presented. It is deemed inclusive that the spatially selected stimulation targets cardiac efferent fibers only, particularly regarding vagal nerve function to organs other than the heart and lungs not assessed (e.g. GI which possesses a large proportion of vagal fibers). In this regard, the title, abstract and many parts in the manuscript overreach the conclusions for selective efferent neuromodulation. It is recommended to concentrate on the identification of a spatial separation of the cardiac afferent and cardiac efferent regions as the important end point. Please revise these parts of the manuscript to reflect this.

We have since modified this according to your and the other reviewer's comments. All changes are highlighted in the manuscript, deleted text was simply removed. We modified the title, rewrote the abstract, edited the introduction and deleted text, and modified the discussion. We hope these revisions now reflect more clearly the important novel findings and goals of the work reported in the manuscript.

2. Please review the journal policy on formatting of a research manuscript: https://jp.msubmit.net/cgi-bin/main.plex?form_type=display_requirements#Revised%20submissions

2a. The abstract must be written to journal policy (see above link). The Journal does not accept a condensed abstract as a substitute for an inadequate full abstract. As written, the background is nearly half of the abstract. This should be reduced to one or two sentences. The purpose or aim of the study must be succinctly and clearly stated. Provide brief description of the research methods and study design. State key techniques. A major part of

the abstract should clearly identify the results of the experiments, expanding and supporting the conclusions drawn in the abstract.

The abstract has been revised. Please see the manuscript document.

2b. Please review the journal policy on reporting animal ethics: <https://physoc.onlinelibrary.wiley.com/hub/animal-experiments>

The Methods section must begin with the subheading "Ethical Approval" wherein authors must state the institutional ethical committee that approved the study and the national guidelines under which the institution operates. The institutional approval code/number should be provided. In addition, the section must include a statement that the investigators understand the ethical principles under which the journal operates and that their work complies with the animal ethics checklist as outlined by the journal.

Identify the source. Detail the supplier or commercial breeder. Specify species, strain, genetic background, weight, sex and age. Indicate the group size and total numbers of animals, including any animals used but subsequently excluded for any reason, specifying any unexpected events. For farm animals it is not necessary to name the commercial supplier. However, information on the country's regulatory authority governing animal husbandry and transportation must be provided. For example: "All animals were reared and transported under conditions specified in the UK's Animal Welfare Act 2006 and The Welfare of Farm Animals (England) Regulations 2007."

State the feeding and water regime e.g. fasting, feeding and water ad libitum or on a specific diet.

We have since added a section title as advised, moved the relevant information up to this section, and added the further required information.

For in vivo experiments under acute surgical anesthesia, please provide an estimate of the total duration of the experiment from induction of anesthesia to endpoint of measurements. An experimental timeline would be very helpful.

The total duration has been added to the paper, as well as an experimental timeline (Figure 2).

Although the manuscript covers the anesthesia protocol in sufficient detail and how the depth of anesthesia was determined and maintained through the experiment, it was not clear when the type of anesthesia was switched from isoflurane to alpha-chloralose. Line 240 states after surgical procedures, but line 271 states "After the initial round of sVNS for identification of pulmonary fascicles, the animal was put back on mechanical ventilation and anesthesia was switched to α -chloralose...". When were the animals switched back to isoflurane, when were they taken off mechanical ventilation? It appears that these statements are conflicting. It would help with clarity if the surgical preparation and experimental procedures were separate sections. Again, an experimental timeline would greatly help.

Thank you for pointing this error/confusion out – we have since corrected this within the text (highlighted) and have added an experimental timeline (now Figure 2) which should provide more clarity to the reader. Additionally, we have shifted some of the sentences from the Surgery section to the Experimental/sVNS sections, renamed and renumbered these sections, and referred to the respective sections within the text where needed. All of this is highlighted within the text.

3. Please review journal policy on statistical reporting: https://jp.msubmit.net/cgi-bin/main.plex?form_type=display_requirements#statistics

The methods section must conclude with a section on data and statistical analysis. It is not clear how variance for means was measured. The Journal requires variance as standard deviation or box whisker plot quartiles. For a given conclusion to be assessed, the exact p values must be stated to three significant figures, unless $p < 0.001$, where $p < 0.001$ is acceptable. Please state samples size and x samples in x animals in text and figure/table legends. Indicate the statistical test used in figures and tables. If uniform throughout, state in the statistical analysis section of methods or how each test was applied to individual experiments.

N numbers have been added throughout the manuscript and figure legends to avoid any confusion. P values have been updated to the exact 3 decimal value. We have additionally added a table (Table 2) from the supplementary data into the manuscript to show all the values reported on for transparency. The statistical tests were uniform throughout – as described in the statistical analysis section – but this has been updated to state this.

4. Supplementary material: The citation of supplementary sheets in the manuscript is very confusing. The supplement obtained is an Excel file that does not identify material as Sheets 1-5. The file is a combination of tables, figures and data with titles but not in any order to identify sheets. And supplement Fig 1 is missing. There is one sheet with just a key. As per Journal policy the figures need to be incorporated into the manuscript. This rule applies also to tables that are needed to interpret results for conclusions. Any remaining supplementary material must be submitted as a PDF file(s). The Excel file also had links to external source(s), that may be blocked by institutional IT systems that prohibit embedded links to external sources (indeed an issue for this editor). Any link to external data must be cited with a full URL to the source, allowing access outside of institutional firewalls.

We have since updated the supplementary files as instructed. The new document will be uploaded upon submission. Additionally, as per the policy, we have added some of the supplementary information into the manuscript where feasible.

REFEREE COMMENTS

Referee #2:

The authors used trial-and-error selective vagal nerve stimulation (sVNS) in vivo in combination with ex vivo micro-computed tomography fascicle tracing and showed a significant spatial separation of cardiac afferent and efferent fibers (179 ± 55 SD microCT, $p < 0.05$ and 200 ± 137 SD, $p < 0.05$ sVNS - degrees of separation across a cross-section of nerve) at the mid-cervical level. They also showed that cardiac afferent fibers are located in proximity to pulmonary fibers, consistent with recent findings of cardiopulmonary convergent neurons and circuits. They demonstrated the ability of sVNS to selectively elicit desired scalable heart rate decrease without stimulating afferent-related reflexes. They concluded that their findings pave the way for more targeted neuromodulation, thereby reducing off-target effects and eliminating the need for titration.

This study was performed by recognized experts in the field of vagus nerve physiology who utilized state of the art methodologies. Its main limitation is the moderate novelty relative to other studies.

Thank you for your comments. We hope to have sufficiently addressed them and believe the edits made to the paper to address the comments have greatly improved the paper and the clarity thereof.

Major comments

1. The authors should refocus their manuscript on the really novel findings (and should adequately highlight them), if any, relative to other studies that have also aimed at defining the spatial distribution and the functional role of cardiac efferent vs afferent vagal nerve fibers. For instance, the recent study by Jayaprakash et al (ref. 21 in the manuscript) has provided a massive amount of anatomical data on efferent vs afferent topography by employing microCT and immunohistochemistry as well.

Thank you for this comment. We have since modified aspects of the introduction (deleted or highlighted changes) and the discussion section to focus more on the novel findings of this work. Please see the manuscript for highlighted changes throughout. With regards to the other studies that have aimed to define spatial distribution and functional organization of the vagus nerve – we have incorporated the Jayaprakash et al manuscript, and have since further elaborated on its findings in the introduction as well as referring to the manuscript by Settell et al (2020). These two studies found that there was a roughly bimodal/half and half distribution of afferent vs efferent fascicles (which was consistent with our findings); however, the study by Settell et al did not trace organ-specific projections and the study by Jayaprakash et al did trace organ-specific projections but not in the same nerves in which afferent and efferent fibers were traced/these were not correlated together. And so, neither of the two studies were able to define, specifically, cardiac efferent vs afferent fibers as the reviewer suggests, which was the main aim of our study presented here.

2. Both in the Introduction and in the Discussion, the authors put much emphasis on the need highly selective VNS for to induce cardioprotection while mitigating off-target effects by specifically targeting pre-ganglionic parasympathetic efferent cardiac fibers. Moreover, they claim a demonstrated ability of sVNS "to selectively elicit desired scalable heart rate decrease". However, the results of functional tests are presented in a single, succinct paragraph on page 18, where the authors report relatively small changes in heart rate without indicating any statistical significance. In addition, there are no data showing selective elicitation of desired scalable heart rate decrease. Once again, what is the difference relative

to previous studies, some of which tested vagal stimulations even in chronically implanted animals? This section of the study needs to be expanded.

The aim of this manuscript was to: determine 1) If there is spatial separation and organization of efferent and afferent cardiac fibers over the cross section of the vagus nerve at the mid-cervical level, and 2) If it is possible to perform sVNS of efferent cardiac fibers whilst avoiding activation of cardiac afferent fibers. The rationale behind this is that it could allow for improved treatment of heart disease by selective activation of cardiac efferent fibers whilst avoiding cardiac afferent, using a multi-electrode nerve cuff, which should reduce side effects and reduce the need for titration when using specific stimulation parameters and selective/targeted stimulation electrodes sufficient for cardiac efferent activation, and not the afferent half of the nerve (where cardiac afferent fibers are located and where it is expected GI fibers would be located – the reason titration is predominantly required). It was not the aim of the paper to decipher the amount of HR drop that would be sufficient for cardioprotection, but rather that we can selectively do so. It was preferential to alter parameters to improve selectivity of HR decrease on the least number of electrodes of the nerve cuff than to alter parameters to get a greater HR decrease on any electrodes. It should be pointed out that in the recent study by Hadaya et al (2023), whole nerve VNS that was delivered at the neural fulcrum (minimal HR change during active VNS), mitigated adverse remodeling of the heart and sympathetic nervous system post-MI, preserved ejection fraction, and reduced the arrhythmia potential. This VNS therapy was delivered 2 days post MI, at a level that reduced HR less than 1-2%, but required a 3-4 week VNS titration prior to MI induction.

Using sVNS, future studies should further optimize the stimulation protocols that elicit the amount of HR decrease sufficient to provide reduction in ventricular arrhythmias and other cardiac symptoms associated with MI and heart disease – all whilst avoiding stimulation of cardiac afferents. We have, however, now elaborated on this in the text (highlighted) within the discussion section, and have referred to previous studies that show that a greater decrease in HR (e.g. 30% as in Radcliffe et al vs 10% in Zhang et al) does not necessarily improve cardiac outcomes but instead a HR decrease from parameters used that are consistent with the neural fulcrum for the individual that can subsequently attenuate heart failure development (Hadaya et al. 2023, Ardell et al. 2017, Premchand et al. 2014).

In the supplementary data, there are tables of the sVNS data for each animal, from each test, showing the electrodes that had the greatest effect, etc. In addition, we have added the raw data tables to the supplementary information from which the HR changes were calculated.

3. Figure 4: it seems these sections were taken at the same level in 5 different animals. The intraneural distribution of the identified fiber types varies notably from animal to animal. Why?

Yes, this is correct – n numbers have been added to the figure legend title to clarify that it is a mid-cervical level cross-section from each of the five animals for which structural imaging was performed. We do not know the exact reason why the intraneural distribution varies from animal to animal but believe the slight variation is similar the variations seen in branching levels or vasculature and muscle variations between animals. In our opinion, however, we believe the distribution is rather similar between the animals and not notable as the reviewer suggests. Looking at what is now figure 5, slight variations can be visualized by eye, but as evident from the data shown in table 1 and section 3.2, as well as the collated data and standard deviations reported in section 3.3, table 2 and figure 7, the composition of the nerves (types and number of fascicles, and fiber-type fascicles) and the overall/average locations were similar between animals with small standard deviations. Cardiac efferent fascicles were almost always located in close proximity to the recurrent laryngeal fascicles, cardiac afferent fascicles were always mixed with pulmonary fibers (and so,

cardiopulmonary) and in close proximity to pulmonary fascicles, the cardiac efferent and cardiac afferent were always located roughly opposite ends of the nerve, and in accordance with above, the pulmonary and laryngeal were split into two halves of the nerve cross-section. In one nerve of 5, the cardiopulmonary fascicles were mixed and not afferent like the other four nerves – but clearly a minority. And, the other variation is the number and placement of fascicles that had merged between the laryngeal and pulmonary fascicles. A line has been added to the discussion to reflect this finding.

Minor comments

1. In Abstract, Introduction and Discussion, the authors write at length about the clinical benefits of sVNS for the treatment of cardiac diseases. These parts of the text can be shortened, as the present study did not test any of such therapeutic interventions. The authors' considerations should instead be more focused on the actual findings and their relevance relative to previous studies.

This was included as it is the rationale as to why we needed to decipher the cardiac-specific afferent and efferent locations. We have since, however, edited this – the abstract, introduction and discussion have been modified in parts to reflect on the main findings of the work.

2. Some parts of the manuscript are repetitive: for instance, lines 173-175 in Introduction and lines 240-242 in Methods, but also others.

Thank you for pointing this out. We have revised the manuscript to remove any repetitions.

Referee #3:

Autonomic imbalance is a significant indicator of adverse clinical outcomes in many cardiovascular diseases. Vagal nerve stimulation has emerged as a popular treatment option for CV diseases. However, the current cervical vagal stimulation devices have off-target side effects due to the lack of cardiac specificity. To address this, Thompson et al. investigated the spatial organisation of the cervical vagus. They tested the feasibility of selective vagal nerve stimulation (sVNS) using a multi-electrode cuff the authors have previously developed.

Thompson and colleagues conducted a study using an anaesthetised pig model, combining functional and paired anatomical investigations. The study's significant strength is the paired data approach. Anatomical micro-CT studies revealed the spatial separation of cardiac afferent and efferent fibres and some spatial specificity of laryngeal and pulmonary fibres. Functional studies demonstrate the feasibility of spatially selective heart rate control; however, cross-activation of other fibres does occur. The study is well designed, given the constraints of achieving such technically challenging experiments. The methods are extensively detailed and very sound; however, some aspects would benefit from clarification. While the study presents compelling data on the potential feasibility of spatially selected vagal nerve stimulation, some conclusions go beyond the data presented and would benefit from additional information.

Major comments:

- The authors present elegant data showing the spatial anatomical location of cardiac, pulmonary and laryngeal-containing fascicles in the cervical vagus but provide no data on the other fibre types present. It would be near impossible to micro-CT the whole vagus. However, providing additional data on the number of efferent and afferent fascicles that are not originating in the neck/thorax, i.e. any fibres that can be traced to the nodose but not to the heart, lungs, or larynx, would give valuable information on just how specific cardiac efferent stimulation could be.

Thank you so much for your comments. There are no fascicles traced from the nodose ganglion down, identifying afferent/efferent, that do not meet with organ-specific fascicles of those traced (heart, lungs, larynx). However, the cardiac efferent fascicles were 100% efferent and 100% cardiac origin. This has been emphasized further in the discussion as the reviewer pointed out in the last comment, this is a major finding of the study. With regards to the others: the fascicles of the thoracic organs, when tracing proximally towards the mid-cervical level of the vagus nerve, merge with other fascicles (that are not from lungs, larynx, or heart). As fibers cannot be traced, fascicles that have merged with another fascicle are labeled as organ-specific fiber containing fascicles depending on which organ is being traced. They merge multiple times along the length but at the cervical level, mostly stay within groups – as reported. Cardiac merges with pulmonary, forming cardiopulmonary and there are a few laryngopulmonary fascicles by the mid-cervical level. If fascicles of a known organ merge with an unknown fascicle, it is labeled as containing fibers from the known organ. We stated that the tracing was done in the same way as our previous study (Thompson et al. 2023) and similar to this study, at the mid-cervical level, there are no unlabeled fascicles and as we traced only thoracic organs, the remaining fibers projecting to the subdiaphragmatic organs must be merged within these fascicles. We hypothesize that the GI fibers are within the mostly afferent side of the nerve. We have added a few lines to the discussion as we acknowledge we did not state this elsewhere – this is highlighted within the manuscript.

- In line with the above, stating that targeted cardiac efferent stimulation can be achieved 'without stimulating afferent reflexes' (Line 52, and others), should say 'cardiac-afferent reflexes'. Other non-cardiac afferent reflexes that are not being recorded from may be

triggered. Moreover, it is difficult to ascertain how sVNS may influence the lungs when the animal is ventilated.

Thank you for this comment. We had clarified this in the majority of the paper but recognize it was not clarified in a few sentences in the abstract and introduction. This has been changed in the manuscript and highlighted.

We have added lines to the discussion to acknowledge the point the reviewer makes in their last sentence above (influence on lungs during ventilation) as well as clarifying that despite the localization of the cardiac afferent and efferent fibers/clusters/fascicles in this study, we did not localize the gastric fibers and thereby cannot confidently state we are able to avoid these too. There is evidence, however, that afferent and efferent is dispersed across the cervical vagus nerve in a mostly bimodal manner, and so GI fibers, which are predominantly afferent, are hypothesized/expected to be located amongst the mostly afferent fascicles. Therefore, they are expected to be located on the opposite side of the nerve to the cardiac efferents which are the target for heart disease treatment.

- The authors focus on how the data presented will 'eliminate the need for titration' and the benefits of achieving faster onset of therapeutic VNS. While this may be a potential benefit of target stimulation, there is no data to support this and stating this as fact is beyond the scope of the study. The authors present data showing that targeted cardiac efferent stimulation will always activate laryngeal fibres - as cough and GI issues are the major side effects that VNS titration aims to avoid the data presented in its current form does not suggest that the need for titration would be 'eliminated'.

Please see the previous comment's response. We have also updated the wording in the manuscript to reflect that we cannot be certain of this – please see the highlighted text in the abstract, introduction and discussion sections; deleted text is not shown. This work is the first step towards achieving this, but it is acknowledged that titration may not be eliminated completely, especially for treating other diseases. We have already acknowledged that the side effects associated with recurrent laryngeal activation may not be avoided due to close proximity with cardiac efferent (target) fascicles.

For the purpose of treating cardiac disease and progression to heart failure post-MI and thereby selectively activating cardiac efferent fascicles, titration could, for the most part, be avoided due to the exclusive nature of the cardiac efferent fascicles, and the bimodal distribution of the mid-cervical vagus nerve (afferent vs efferent as mentioned above) when using a multi-electrode, selective nerve cuff.

Minor comments:

Introduction:

- Citing some of the original cardiac VNS literature (e.g. Sunagawa's Circulation paper from 2003) would be beneficial, instead of just citing articles from the research team.

We have since added a number of other references to the introduction. This is highlighted within the updated manuscript.

- Introduction is, at times, repetitive and could be shortened.

We have removed repetitions throughout the paper and have edited the introduction in line with comments from the editors and reviewers. Please see the manuscript with the new version.

Methods:

- How many pigs were done in each location? Was there any failure rate in the functional studies?

There was only one location (UCLA Cardiac Arrhythmia Center) and all 10 animal experiments were performed there. This was performed in two batches – i.e. two trips of the UCL team to the UCLA center. Each trip consisted of 6 experiments each. The first pig of each trip was used for testing of all the equipment and surgery setup with new surgeons, etc., and were excluded from the study. In the first trip, spontaneous breathing was not performed, and so, pulmonary testing was not performed in 5 of 10 animals. This has now been clarified in the methods section 2.4.2. Additionally, added to this section is the numbers that were successful of the functional studies (one pulmonary testing did not work and one recurrent laryngeal testing did not work – the EMG needles did not pick up clear, noise-free activity). Thank you for pointing this out – it was clear in the supplementary tables but we had not clarified in the text and are grateful that this was noticed before publication.

- Clarifying how many studies were paired, n = 5? and how many were functional-only studies would help transparency.

We have since made this clear in the respective methods sections. Please see highlighted text.

- The experimental timeline paragraph (lines 210-221) could be better presented as a timeline.

We have since added an experimental timeline (Figure 2).

- Why were there stimulation electrodes on the recurrent laryngeal nerve?

Thank you for pointing this out. We have since removed this sentence and the sentence on EIT (electrical impedance tomography for which the recurrent laryngeal stimulation electrode was used for) as this was not reported on/included in the study. This correlates with the purple section in the new experimental timeline figure (Figure 2).

- When was fentanyl stopped?

Fentanyl was used to maintain analgesia throughout the experiment and was used at the discretion of the surgeons/anesthetists. It was stopped completely upon euthanasia (please see the experimental timeline figure for clarity)

- Was the left vagus also stimulated or recorded during the procedure (line 258)? Direct recording from the left vagus after right vagal afferent stimulation would be interesting.

In line with the comment two above this one, thank you for pointing this out and the sentence about adding the left vagus nerve cuff was removed from the manuscript. This, again, was used for EIT and not included in this manuscript.

- Were the pulmonary responses tested under isoflurane anaesthesia? How many Ephys animals underwent pulmonary testing? This appears low in the supplemental data (thank you for the extensive supplemental data)- The addition of specific n to figures would help transparency.

Yes, the pulmonary responses were tested under isoflurane anesthesia. The anesthesia was then transitioned to alpha-chloralose for the remainder of the in vivo experiment. This should be clear in the experimental figure that we have since added. Regarding the n numbers for the respective organs/tests, please see the response to your first methods comment.

Results:

- With data presented as per cent change, please provide a table of raw data values.

We have since added the tables of raw data values to the supplementary information. This is referred to in the text – first paragraph of section 3.1 where the percent changes are reported.

- Figure 3. A great raw data figure, but some aspects do not match what is written in the methods. On/off periods are stated to be 15 seconds for everything but laryngeal stimulation, but panel D shows something different. Heart rates are lower than what is stated as the target range in the methods. Why are some electrode combinations missing from the afferent stimulation?

Thank you for pointing out these discrepancies. The on/off period for the cardiac stimulation post-vagotomy has been clarified in the text. The reasoning for this has also been added to the discussion section, where this time period was mentioned but has now been further elaborated on and highlighted. We have also added the stimulation times to the figure description so it can be clear. Thank you for pointing out the HR differences – this was a typo within the methods section and has since been updated. For the afferent stimulation, only every 2nd pair of electrodes (so, 7/14 pairs) was used for stimulation. This was out of the interest of time as it was at the end of the experimental day and the allotted experimental time, multiple parameters were being tested, and as above, due possibly to the reflex activation via afferents to the other vagus nerve or the sympathetic pathways, there was a delay in response to stimulation and so sufficient time of (instead of the 15 secs off) up to a minute was incorporated between on periods, to allow for this delayed response and corresponding change in HR and for return to normal functioning/baseline activity.

Discussion

- Rationale for using the right VNS.

This has been added to the end of the first paragraph (highlighted) in the discussion.

- Line 469: The number of cardiac efferent fascicles is low, but if they are 100% efferent when originating from the cardiac vagal branch, this is a significant observation! And could help develop targeted VNS strategies.

Thank you for this comment. We agree – and this is in line with the purpose and rationale of our study. We have now emphasized this in the discussion, in conjunction with clarifying and refocusing the manuscript on the main, novel findings of the study, as suggested by the editors and reviewers.

END OF COMMENTS

Dear Dr Thompson,

Re: JP-RP-2024-286494R1 "Towards spatially selective efferent neuromodulation: Anatomical and functional organization of cardiac fibers in the porcine cervical vagus nerve" by Nicole Thompson, Enrico Ravagli, Svetlana Mastitskaya, Ronald Challita, Joseph Hadaya, Francesco Iacoviello, Ahmad Shah Idil, Paul R Shearing, Olujimi A Ajjola, Jeffrey Laurence Ardell, Kalyanam Shivkumar, David Holder, and Kirill Aristovich

Thank you for submitting your manuscript to The Journal of Physiology. It has been assessed by a Reviewing Editor and by 2 expert referees and we are pleased to tell you that it is acceptable for publication following satisfactory revision.

REVISION CHECKLIST:

We look forward to receiving your revised submission.

Yours sincerely,

Harold Schultz
Senior Editor
The Journal of Physiology

EDITOR COMMENTS

Reviewing Editor:

Thank you for carefully going through and addressing the comments made by all reviewers. I think the changes made will greatly improve the accessibility and clarity of the work.

Senior Editor:

Thank you for submission of your revised research article to the Journal of Physiology for consideration. The revision has been reviewed by the original referees and found to be acceptable for publication with a remaining suggestion from referee 1 and a couple of remaining concerns from the senior editor. Please address these concerns.

1. The editors agree that the final 4 sentences of the abstract are simply a general (and redundant) discussion of relevance and the space could be better used to describe more of the important details of the outcomes, as summarized in the first paragraph of the discussion. Also please state the animal model studied.

2. In Table 1 of the manuscript, please define the color coding.

3. The supplementary file needs to be a PDF document for processing. Also legends are needed in several of the tables to understand abbreviations and what is reported. Tables starting on page 6, please describe what is being reported and a color key. Starting on page 12, please label the columns and what is being reported. The color shading in column 2 is not intuitive, please describe. Starting on page 20 for nerve 6 -10, please describe what is reported in the tables and the color coding.

REFEREE COMMENTS

Referee #2:

The authors were very responsive and their manuscript is now better focused and organized. One final request: the last 4 (long) sentences of the abstract are speculative take a lot of words. They can be condensed in one or two sentences and the saved spaces can be taken by additional information strictly related to the results.

Referee #3:

The Authors have adequately responded to all my comments.

END OF COMMENTS

1st Confidential Review

28-Jun-2024

Responses

EDITOR COMMENTS

Reviewing Editor:

Thank you for carefully going through and addressing the comments made by all reviewers. I think the changes made will greatly improve the accessibility and clarity of the work.

Thank you. We agree the comments and subsequent changes greatly improved the paper.

Senior Editor:

Thank you for submission of your revised research article to the Journal of Physiology for consideration. The revision has been reviewed by the original referees and found to be acceptable for publication with a remaining suggestion from referee 1 and a couple of remaining concerns from the senior editor. Please address these concerns.

Thank you. We have since made the final changes to the paper and supplementary file.

1. The editors agree that the final 4 sentences of the abstract are simply a general (and redundant) discussion of relevance and the space could be better used to describe more of the important details of the outcomes, as summarized in the first paragraph of the discussion. Also please state the animal model studied.

We have changed the end of the abstract as specified above. Please see the highlighted changes in the manuscript.

2. In Table 1 of the manuscript, please define the color coding.

Apologies for not including this previously. This has been added to the table description. Please see the highlighted changes.

3. The supplementary file needs to be a PDF document for processing. Also legends are needed in several of the tables to understand abbreviations and what is reported. Tables starting on page 6, please describe what is being reported and a color key. Starting on page 12, please label the columns and what is being reported. The color shading in column 2 is not intuitive, please describe. Starting on page 20 for nerve 6 -10, please describe what is reported in the tables and the color coding.

We have provided more detail in the figure and table legends for all the supplementary information – explaining what colors mean, what is shown in each table, what the values are, differences between tables (e.g. normalized to 1), etc. Additional column titles have been

added for those columns that were not previously labeled. Color shading has either been explained before tables shown, or removed if not relevant. This document has been made into a pdf for the submission.

REFEREE COMMENTS

Referee #2:

The authors were very responsive and their manuscript is now better focused and organized. One final request: the last 4 (long) sentences of the abstract are speculative take a lot of words. They can be condensed in one or two sentences and the saved spaces can be taken by additional information strictly related to the results.

The abstract has been edited as specified by the reviewer and senior editor. Please see the highlighted changes.

Referee #3:

The Authors have adequately responded to all my comments.

Thank you.

END OF COMMENTS

Dear Dr Thompson,

Re: JP-RP-2024-286494R2 "Towards spatially selective efferent neuromodulation: Anatomical and functional organization of cardiac fibers in the porcine cervical vagus nerve" by Nicole Thompson, Enrico Ravagli, Svetlana Mastitskaya, Ronald Challita, Joseph Hadaya, Francesco Iacoviello, Ahmad Shah Idil, Paul R Shearing, Olujimi A Ajjola, Jeffrey Laurence Ardell, Kalyanam Shivkumar, David Holder, and Kirill Aristovich

Thank you for submitting your manuscript to The Journal of Physiology. It has been assessed by a Reviewing Editor and we are pleased to tell you that it is acceptable for publication following satisfactory revision.

REVISION CHECKLIST:

We look forward to receiving your revised submission.

Yours sincerely,

Harold Schultz
Senior Editor
The Journal of Physiology

EDITOR COMMENTS

Thank you for submission of your revised research article to the Journal of Physiology for consideration. The information in the Supplementary tables still is not sufficiently explained for the reader to clearly understand.

For the 10 tables starting on page 7, it is not clear which are laryngeal EMG, pulmonary EtCO₂, or cardiac heart rate, and units for the values are not provided. Please define what is meant by 'response from baseline'? Does this mean change from baseline.

For the 10 tables starting on page 12, it is not clear which are laryngeal EMG, pulmonary EtCO₂, or cardiac heart rate.

For the MicroCT per nerve measurements starting on page 17, in the legend, please define the subsequent illustrations as Fig 1, Table 1, etc. And for each subsequent nerve group, provide a title to each illustration, referencing what is described from the legend (e.g. Fig 1, Fig 2: Table 1, Table 2 etc.)

We can proceed to acceptance once these concerns are addressed.

END OF COMMENTS

2nd Confidential Review

06-Aug-2024

Responses

EDITOR COMMENTS

Thank you for submission of your revised research article to the Journal of Physiology for consideration. The information in the Supplementary tables still is not sufficiently explained for the reader to clearly understand.

Apologies for not clarifying the supplementary data sufficiently. We hope the new version is much clearer now as we have incorporated all the suggested changes.

For the 10 tables starting on page 7, it is not clear which are laryngeal EMG, pulmonary EtCO₂, or cardiac heart rate, and units for the values are not provided. Please define what is meant by 'response from baseline'? Does this mean change from baseline.

Thank you for pointing this out as it definitely needed further clarification and rectification within the document. We have since edited this to explain in the sentences prior to the 10 tables more clearly, and have added a row in each table to provide more details about each measurement/data and its unit:

"In each of the following 10 tables (Tables 3.1 - 3.10), the relative variation between stimulation on and stimulation off ($(Val(stim)-Val(rest))/Val(rest)$) for each of the respective readings (laryngeal EMG, pulmonary respiratory rate, cardiac heart rate) is shown for the recurrent laryngeal function (L), pulmonary function (P), cardiac efferent function (pre-vagotomy, C) and cardiac afferent function (post-vagotomy, CP)."

e.g. table:

Supplementary Table 3.1

Nerve

6

Electrodes	Fascicle Type			
	'L'	'P'	'C'	'CP'
	Relative variation of EMG (μV) between stim on and stim off [unitless]	Relative variation of RR (breaths per minute) between stim on and stim off [unitless]	Relative variation of HR (beats per minute) between stim on and stim off [unitless]	Relative variation of HR (bpm) between stim on and stim off [unitless]
E3	5.28		-0.12	0.14
E4	0		0.01	0.14
E5	-0.01		0	0.2
E6	-0.01		0.01	0.2
E7	0		0.01	0.21
E8	0.03		0	0.21
E9	0.02		0	0.16
E10	0		0.01	0.16
E11	675.9		0.01	0.16
E12	649.32		0	0.16
E13	79.7		-0.09	0.16
E14	56.05		-0.09	0.16
E1	5.97		-0.08	0.04
E2	6.16		-0.09	0.04
max	675.9	0	0.12	0.21

For the 10 tables starting on page 12, it is not clear which are laryngeal EMG, pulmonary EtCO₂, or cardiac heart rate.

As above, further clarification was given before the tables are presented and rows were added to each table to describe the data type.

For the MicroCT per nerve measurements starting on page 17, in the legend, please define the subsequent illustrations as Fig 1, Table 1, etc. And for each subsequent nerve group, provide a title to each illustration, referencing what is described from the legend (e.g. Fig 1, Fig 2: Table 1, Table 2 etc.)

We have now labelled each Section (section 1, 2, 3, and 4) within the supplementary document to allow for accurate, consecutive table and figure numbers throughout. Therefore, we have given table numbers to all the sVNS data prior to the microCT data, and have subsequently given microCT data within section 4 figure and table numbers e.g. 4.1.1 and 4.5.6, etc. Each table and figure has a description of what is shown/reported and for which nerve.

We can proceed to acceptance once these concerns are addressed.

Thank you so much. We hope we have sufficiently rectified the supplementary document and provided enough clarification.

END OF COMMENTS

Dear Dr Thompson,

Re: JP-RP-2024-286494R3 "Towards spatially selective efferent neuromodulation: Anatomical and functional organization of cardiac fibers in the porcine cervical vagus nerve" by Nicole Thompson, Enrico Ravagli, Svetlana Mastitskaya, Ronald Challita, Joseph Hadaya, Francesco Iacoviello, Ahmad Shah Idil, Paul R Shearing, Olujimi A Ajjola, Jeffrey Laurence Ardell, Kalyanam Shivkumar, David Holder, and Kirill Aristovich

We are pleased to tell you that your paper has been accepted for publication in The Journal of Physiology.

Authors should note that it is too late at this point to offer corrections prior to proofing. Major corrections at proof stage, such as changes to figures, will be referred to the Editors for approval before they can be incorporated. Only minor changes, such as to style and consistency, should be made at proof stage. Changes that need to be made after proof stage will usually require a formal correction notice.

If you would like to receive our 'Research Roundup', a monthly newsletter highlighting the cutting-edge research published in The Physiological Society's family of journals (The Journal of Physiology, Experimental Physiology and Physiological Reports), please click this link, fill in your name and email address and select 'Research Roundup': <https://www.physoc.org/journals-and-media/membernews/>.

Yours sincerely,

Harold Schultz
Senior Editor
The Journal of Physiology

P.S. - You can help your research get the attention it deserves! Check out Wiley's free Promotion Guide for best-practice recommendations for promoting your work at www.wileyauthors.com/eeo/guide. You can learn more about Wiley Editing Services which offers professional video, design, and writing services to create shareable video abstracts, infographics, conference posters, lay summaries, and research news stories for your research at www.wileyauthors.com/eeo/promotion.

IMPORTANT NOTICE ABOUT OPEN ACCESS: To assist authors whose funding agencies mandate public access to published research findings sooner than 12 months after publication, The Journal of Physiology allows authors to pay an Open Access (OA) fee to have their papers made freely available immediately on publication.

You can check if your funder or institution has a Wiley Open Access Account here: <https://authorservices.wiley.com/author-resources/Journal-Authors/licensing-and-open-access/open-access/author-compliance-tool.html>.

EDITOR COMMENTS

Senior Editor:

The editors wish to thank the authors for these final adjustments to the manuscript. The article is now accepted for publication. Congratulations for an interesting and insightful study. Please consider the Journal of Physiology for your future studies.